# A Simulation and Optimization Study of the Swirling Nozzle for Eccentric Flow Fields of Round Molds

**Peng Lin, Yan Jin \*** , **Fu Yang, Ziyu Liu, Rundong Jing, Yang Cao, Yuyang Xiang, Changgui Cheng and Yang Li**

Key Lab for Ferrous Metallurgy and Resources Utilization of Ministry of Education, Wuhan University of Science and Technology, Wuhan 430081, China; womenshidage@163.com (P.L.); yangfu9402@163.com (F.Y.); liuziyux@sina.com (Z.L.); jrd1754268947@163.com (R.J.); yangcao060@gmail.com (Y.C.); xiangdashuai2020@163.com (Y.X.); ccghlx@263.net (C.C.); liyang@wust.edu.cn (Y.L.)

**\*** Correspondence: jinyan@wust.edu.cn; Tel.: +86-156-9718-0966

**Abstract:** In continuous casting, the nozzle position may deviate from the center under actual operating conditions, which may cause periodic fluctuation of the steel-slag interface and easily lead to slag entrapment and gas absorption. Swirling nozzles can reduce these negative effects. A mathematical simulation method based on a round mold of steel components with a 600 mm diameter is applied to study the flow field of molten steel in a mold. The swirling nozzle is optimized through the establishment of a fluid dynamics model. Meanwhile, a 1:2 hydraulic model is established for validation experiments. The results show that, when the submerged entry nozzle (SEN) is eccentric in the mold, it results in serious bias flow, increasing the drift index in the mold up to 0.46 at the eccentric distance of 50 mm. The impact depth of liquid steel and turbulent kinetic energy can be decreased by increasing the rotation angle of the nozzle. The nozzle with one bottom hole, which significantly decreases the bottom pressure and turbulent kinetic energy, greatly weakens the scour on nozzle and surface fluctuation. In the eccentric casting condition, using the optimized swirling nozzle that employs a 5-fractional structure, in which the rotation angle of 4 side holes is 30° and there is one bottom outlet, can effectively restrain bias flow and reduce the drift index to 0.28, a decline of more than 39%.

**Keywords:** round mold; swirling nozzle; flow field; eccentric casting; mathematical simulation

---

## 1. Introduction

The mold is the key factor in quality control of casting molding. The molten steel flow in a mold directly effects thickness and uniformity of the solidified shell [1,2], which is a complex turbulent flow process. Its main characteristics are irregularity, rotation, three-dimensionality, diffusivity and dissipativity [3,4]. As the inlet of the mold, the outlet of the nozzle influences the flow pattern of liquid steel directly in the mold and affects the flow field and liquid surface behavior of steel slag [5–8]. Therefore, it is very important for the shape and control of the flow field in the mold. According to the actual operation of a steel works with high productivity and low cost, a tundish pours the round strands with diameters of 500 mm and 600 mm at the same time. This causes the submerged entry nozzle of a 600 mm mold to deviate from the center by 50 mm during the casting process. The operation results in uneven distribution of the flow field, a large surface wave and uneven thickness of the solidified shell [9,10]. It was found that the swirling flow field driven by swirling nozzle could inhibit the asymmetrical flow in mould and could suppress the negative effects on the surface and/or internal quality of strand caused by asymmetrical flow [6,7]. Based on previous work [9–12] and through the study of numerical simulation and water modeling on the eccentric flow field of the mold, in this paper,

a reasonable swirling-type submerged entry nozzle is designed, and the swirling nozzle is adopted to optimize the flow field in the mold to improve the casting environment and enhance the quality of the round bloom.

## 2. Mathematical and Physical Models

### 2.1. The Research Subject

The subject investigated in this study is a round mold in a caster, which is used in conjunction with a 46 t four-flow tundish. To research the flow state and distribution of molten steel in the mold of different nozzle structures under the conditions of different eccentric degrees on casting, the original five-fractional radial nozzle (the original SEN) and swirling five-fractional tangential nozzle (the swirling SEN) are adopted. The SEN structure and parameters are shown in Figure 1 and Table 1.

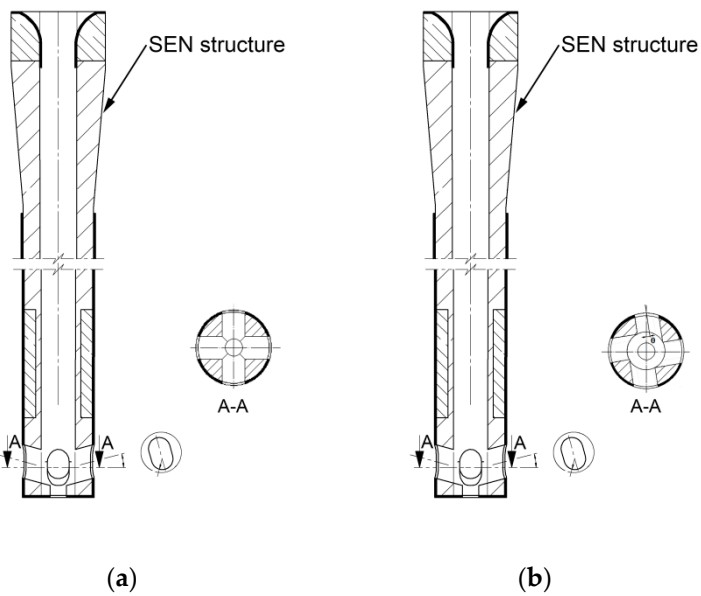

| (a) | (b) |

**Figure 1.** Structural schematic of the mold and submerged entry nozzle. (**a**) Five- fractional SEN with outlets in radial direction (**b**) Five- fractional SEN with outlets in tangential direction.

**Table 1.** Geometric parameters of various SENs.

| Nozzle Type | Inside Diameter (mm) | Outside Diameter (mm) | Tangential Angle (°) | Side Hole Height (mm) | Side Hole Width (mm) |
|---|---|---|---|---|---|
| Five-fractional radial nozzle (a) | 50 | 100 | 0 | 48 | 32 |
| Five-fractional tangential nozzle (b) | 50 | 100 | 15, 30 | 48 | 32 |

### 2.2. The Mathematical Model

#### 2.2.1. Model Assumptions

According to the physical characteristics of liquid steel and its flow features, the following assumptions have been created:

(1) Neglecting the influence of a surface covering agent on the flow, the top surface of the molten steel is a free-slip surface that is satisfied with $k = \varepsilon = 0$ and $\partial u/\partial z = \partial v/\partial z = \partial k/\partial z = \partial \varepsilon/\partial z = w = 0$;

(2) The fluid in the mold is a viscous and incompressible fluid, which has unsteady flow and the initial temperature is a uniform distribution;

(3) The effects of the shrinkage of the round bloom and the vibration of the mold on the flow of molten steel are not considered when the steel solidifies;

(4)     The latent heat of $\delta$-$\gamma$ phase transformation is far less than that of solidification, neglecting the effect of the $\gamma$ metal solid phase transformation and the influence of solidification of the billet shell on the molten steel flow in the mold;

(5)     Argon bubbles and inclusions are spherical, without considering collision and growth;

(6)     The inclusions do not influence the flow field, but the flow field does influence the movement of the inclusions.

### 2.2.2. The Governing Equations and Boundary Conditions

As a complex process, the flow behavior and temperature change of molten steel have been simulated by the continuity equation, the momentum equation (the Navier-Stokes equation), the $k$-$\varepsilon$ two equation describing the turbulence model and energy equation (coupled with momentum and continuity equations). The Discrete Phase Model (DPM) is established to simulate the trajectories and distributions of the inclusions and argon bubbles in the mold:

(1)     The continuity equation:

$$\partial\rho/\partial t + \partial\rho u_j/\partial x_j = 0 \tag{1}$$

In the formula, $u_j$ represents the velocity component (m/s); $x_j$ represents coordinate (m); $\rho$ and $t$ are fluid density (kg/m$^3$) and time (s), respectively;

(2)     The Navier-Stokes equation:

$$\partial(\rho u_i)/\partial t + \partial(\rho u_i u_j)/\partial x_j = -\partial p/\partial x_i + \partial[\mu_{\text{eff}}(\partial u_i/\partial x_j + \partial u_j/\partial x_i)]/\partial x_i + \rho g_i \tag{2}$$

where $u_i$ and $u_j$ denote the velocity of i and j direction (m/s); $x_i$ and $x_j$ stand for coordinates of the i and j directions (m); $p$ is pressure (Pa); $\mu_{\text{eff}}$ indicates the coefficient of effective viscosity (Pa·s) determined by the available turbulence model; and $g_i$ stands for the gravity component, (m/s$^2$);

(3)     The $k$-$\varepsilon$ two equation model. The $k$ equation of turbulent kinetic energy is given by the expression:

$$\partial(\rho k)/\partial t - \partial(\rho k u_j)/\partial x_j = \partial[(\mu_t/\sigma_k - \mu)\,(\partial k/\partial x_i)]/\partial x_j + \mu_t\,(\partial u_j/\partial x_i)\,(\partial u_i/\partial x_j + \partial u_j/\partial x_i) - \rho\varepsilon \tag{3}$$

In this formula, $k$ represents turbulent kinetic energy (m$^2$/s$^2$), and $\varepsilon$ denotes the dissipation rate of turbulent flow energy (m$^2$/s$^3$); The $\varepsilon$ equation of dissipation rate of turbulent energy:

$$\partial(\rho\varepsilon)/\partial t + \partial(\rho\varepsilon u_j)/\partial x_j = \partial[(\mu_t/\sigma_\varepsilon - \mu)\,(\partial\varepsilon/\partial x_i)]/\partial x_i + C_1(\varepsilon/k)\mu_t(\partial u_j/\partial x_i)(\partial u_i/\partial x_j + \partial u_j/\partial x_i) - C_2\rho(\varepsilon^2/k) \tag{4}$$

where $\mu_t$ can be calculated by Equations $\mu_t = \rho C_\mu k^2/\varepsilon$ and $\mu_{\text{eff}} = \mu + \mu_t$; in the formula, $\mu_t$ represents the turbulent viscosity (Pa·s); $\mu$ is the laminar viscosity (Pa·s); $C_1$, $C_2$, $C_\mu$, $\sigma_\varepsilon$ and $\sigma_k$ are empirical constants, $C_1 = 1.43$, $C_2 = 1.93$, $C_\mu = 0.09$, $\sigma_k = 1.0$, $\sigma_\varepsilon = 1.43$.

(4)     The energy transfer equation. The energy transfer equation was coupled with the flow field equations. Without considering the effect of temperature on the density of molten steel, the flow field and temperature field of the mold are calculated by single-phase coupling. The energy transfer equation is shown below:

$$\partial(\rho u_j H)/\partial x_i = \partial[k_{\text{eff}}(\partial T/\partial x_i)]/\partial x_i \tag{5}$$

where keff can be calculated by equation keff = $\mu/\sigma T + \mu t/\sigma t, T$; in the formula, H represents the specific enthalpy of molten steel (J/kg); $T$ stands for molten steel temperature (K); and $k_{\text{eff}}$ is the effective thermal conductivity (W/m/K); $\sigma_T = 1.00$ (K·s$^2$/m$^2$); $\sigma_{t,T} = 0.9$ (K·s$^2$/m$^2$).

(5) The DPM Model. In the integral Lagrangian coordinate system, the trajectory of a particle (an inclusion or an argon bubble) is described by the particle force differential equation:

$$\partial u_p / \partial t = F_D (u - u_p) + g_i (\rho_p - \rho)/\rho_p + F_i \tag{6}$$

$$F_D = \left(18\mu_i / \rho_p d_p^2\right)(C_D Re/24) \tag{7}$$

In the formula, $u$ and $u_p$ are the velocity of the fluid phase and the particle phase (m/s); $\mu_i$ is the dynamic viscosity of the fluid (Pa·s); $\rho$ and $\rho_p$ are the density of the fluid phase and the particle phase (kg/m$^3$); $d_p$ is the particle diameter (m); Re is the Reynolds number of the particle; $F_D$ is the momentum exchange coefficient of drag force (1/s); $F_i$ is the liquid inertia force per particle mass acting on particle as particle accelerating (i = {x, y, z}) (m/s$^2$); $C_D$ is the drag force coefficient (-).

In the model, the range of particle size for simulated inclusions is 1~50 μm. The density of inclusions is 3000 kg/m$^3$, and the diameter distribution of particles is discontinuous and is divided into 11 sizes, including 1, 5, 10, 15, 20, 25, 30, 35, 40, 45 and 50 μm. For the simulation of argon bubbles, the drag force, buoyancy force and virtual mass force on bubbles are considered in calculation. The density of argon is 0.275 kg/m$^3$, and the particle size range of argon bubbles is 0.6–3.0 mm and its mean diameter is 2 mm according to the previous experiments. The particle size distribution was set as Rosin-Rammler distribution, in which the different bubble size ranges are divided into discrete size groups, as shown in Equation (8):

$$Y_d = \exp\left[-(d_p/d_{a,avg})^n\right] \tag{8}$$

where $Y_d$ is the mass fraction of particles; $d_{a,avg}$ is the mean bubble diameter (m); n is the distribution index.

Boundary conditions of the model have been summarized as follows:

(1) The entrance is defined at the inlet of liquid steel in the upper part of the submerged nozzle, entrance velocity ($V_{in}$) is determined by casting speed in steady pouring according to the principle of mass conservation of molten steel; the temperature of molten steel at the entrance is 1803 K; the turbulent kinetic energy ($k_{inlet}$) and turbulent energy dissipation rate ($\varepsilon_{inlet}$) at the entrance are calculated by $k_{inlet} = (3/2)(V_{in}T_i)^2$ and $\varepsilon_{inlet} = C_\mu^{3/4}k^{3/2}/l$ respectively, in which $T_i$ of the turbulence intensity is 3.7%, and $l$ of the turbulence length scale is 0.07$d$, among which $d$ is nozzle diameter; and $C_\mu$ is taken as 0.09;

(2) The exit is set up at the bottom of the calculation domain as free exit "outflow";

(3) The liquid surface of the mold is set as a free surface, and the upper surface temperature of the liquid slag layer is a constant temperature of 1500 K, with the velocity component perpendicular to the liquid surface of a constant zero;

(4) The wall of the mold with a no-slip surface is treated by the standard wall function of Fluent, and the height distance from the meniscus level dependant heat flux distributionon with the average heat flux (745 kW/m$^2$) is employed as the heat transfer condition.

When using a discrete phase model to simulate the process of argon blowing or inclusion movement, argon bubbles blow into the mold from the submerged nozzle with the flow rate of 2 L/min. For argon bubbles, the boundary condition of the outlet of mold is set to "escape" condition, the liquid surface is set to "trap" condition, and the wall of the mold is set to "reflect" boundary condition. The inclusions are injected into the inlet at the same velocity as that of the molten steel, and the diameter distribution of inclusions is set to "uniform", which means that the number of inclusion particles for each diameter is the same. The inclusions can be trapped at the liquid surface and reflected by the wall of mold, and the model outlet is the escape outlet.

According to the assumptions, Equations and boundary conditions of the model, a geometric model of the mold is established using ICEM software (15.0, Ansys Inc., Canonsburg, PA, USA) to carry on the grid division. The meshes of the mold computational domains included non-uniform

grids with about 1.55 million cells, and the maximum mesh size is set to 16mm. In order to improve the calculation accuracy of complex structures of nozzle, the nozzle area is encrypted with the maximum grid size of only 8mm. Finally, the mathematical model is set up in FLUENT software (ver. 15.0, Ansys Inc.) to complete the calculation. The unsteady state calculation is employed to simulate molten steel flow with the time step of 0.005 s. To ensure the development of the flow field in the mold, the calculation domain of the model has also been extended to twice the actual mold length. The process parameters and thermal physical properties are shown in Tables 2 and 3 [12]. Among them, the immersion depth of the nozzle is defined as the distance from the upper end surface of the SEN side hole to the surface of molten steel. The numerical model widely applies the SIMPLE algorithm by utilizing commercial software to simulate the three-dimensional flow field and temperature field in the mold. The movement track of inclusion particles or bubbles in the mold is simulated by Coupled algorithm. The residual value equation is less than $10^{-4}$ of the convergence standard solving the continuity equation, momentum equation and turbulence. The process of building the FLUENT model is shown in Figure 2.

**Table 2.** Parameters for the mold.

| Section Diameter /(mm) | Liquid Surface of Stable Casting (mm) | Liquid Surface of Overflow (mm) | Immersion Depth of Nozzle (mm) | Casting Speed (m/min) | Casting Temperature (K) |
|---|---|---|---|---|---|
| 600 | 900 | 950 | 160 | 0.30 | 1803 |

**Table 3.** Thermal physical properties of molten steel.

| Density (kg/m$^3$) | Viscosity (Pa·s) | Molar Mass (g/mol) | Specific Heat Capacity (J/(kg·K)) | Effective Thermal Conductivity (W/(m$^2$·K)) |
|---|---|---|---|---|
| 7000 | 0.0065 | 55.85 | 750 | 41 |

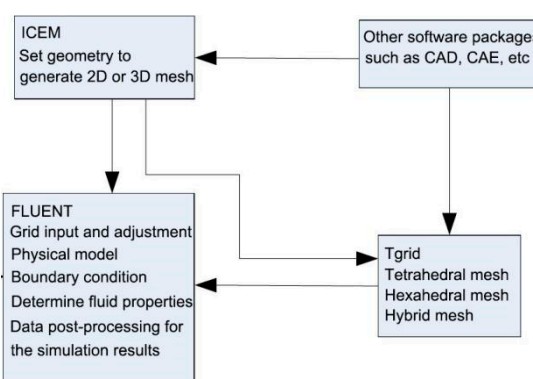

**Figure 2.** The process of building the FLUENT model.

### 2.3. The Physical Model

According to the principle of similarity, in order to ensure the similarity of fluid motion behavior between the model and prototype, and to consider the effects of inertia force, gravity, surface tension and viscous force of the melt, it is necessary that the Reynolds, Froude, and Webber numbers for the prototype and model are equal, and the relevant calculation formulas are detailed in Equations (9), (10) and (11). According to the Equation (9) used for the calculation of the Reynolds number, the Reynolds numbers of the model and the prototype are by far more than $10^5$, thus belonging to the second self-modelling zone. Because the flow is in the same self-modeling zone, the velocity distribution of the fluid is not affected by Reynolds number, and the viscous force can be ignored in the establishment of the water model. Therefore, the similarity of the model and prototype can be ensured when the Froude and Weber numbers are equal, which does not require that the Reynolds number be equal. Considering the accuracy of the experiment and the experimental conditions, the similarity ratio of 1:2 between the model and the prototype is adopted to build physical model by applying plexiglass

materials, as shown in Figure 3. The specific parameters of the model and the prototype are detailed in Table 4.

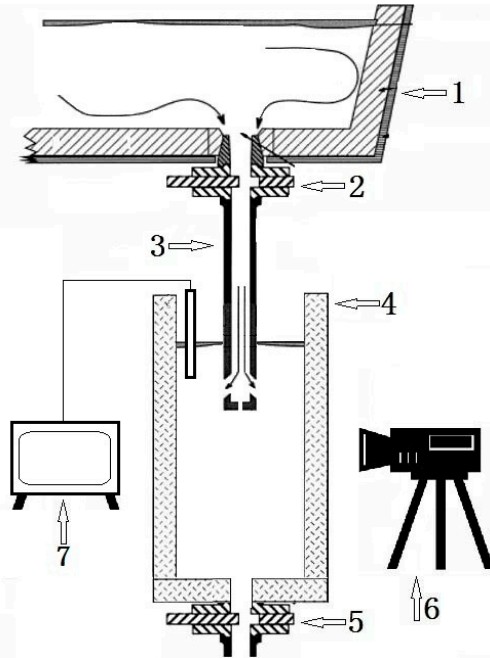

**Figure 3.** Water modeling experiment schematic. (1) Casting Tundish; (2), (5) slide gate; (3) Submerged nozzle; (4) Mold; (6) High speed camera; (7) DJ800 hydraulic acquisition treatment system.

**Table 4.** Water model experiment parameters.

| Parameters | Prototype | Model |
|---|---|---|
| Section diameter (mm) | 600 | 300 |
| Casting speed (m/min) | 0.3 | 0.212 |
| Water flow speed (m$^3$/h) | 5.09 | 0.900 |
| Mold length (mm) | 900 | 1000 |
| Immersion depth (mm) | 160 | 80 |
| Eccentric distance of SEN (mm) | 50 | 25 |
| Diameter of SEN (mm) | 50 | 25 |
| Steel- slag interfacial tension coefficient (N/m) | 1.4 | - |
| Water- oil interfacial tension coefficient (N/m) | - | 0.05 |
| Reynolds number | $2.65 \times 10^8$ | $1.27 \times 10^7$ |
| Froude number | 1.06 | 1.06 |
| Weber number | $1.30 \times 10^2$ | $1.30 \times 10^2$ |

Molten steel and the specific viscosity of the slag are simulated by water and the proper proportion of the mixed oil. The inflow and outflow of water are adjusted by the rotameter to simulate the different speeds. The height of liquid surface is collected by a DJ800 (China Institute of Water Resources and Hydropower Research, Beijing, China) acquisition and processing system, and the measuring range and accuracy of the wave-height sensor are 15 cm and 0.5% full scale (F.S.). Methylene blue is employed as a tracer dye on the flow field in the mold through camera equipment to monitor and record the distribution and development of the tracer in various technological conditions:

$$Re = u_o d_o / v \tag{9}$$

$$Fr = u_o^2 / g d_o \tag{10}$$

$$We = \rho u_o^2 d_o / \sigma \tag{11}$$

In these formulas, Re is the Reynolds number; Fr is the Froude number; We is the Weber number; $u_o$ is the liquid flow velocity of the SEN outlet (m/s); $d_o$ is the inside diameter of the SEN (m); $v$ is the kinematic viscosity of liquid (m$^2$/s); σ is the interfacial tension coefficient (N/m) [13].

After determining the proportion of water model parameters, it is necessary to determine the similar conditions of steel slag interface. The dimension analysis shows that in order to simulate the flow of steel slag interface, the medium used to simulate the mold powder should satisfy the following condition [11]:

$$v_{slag}/v_{steel} = v_1/v_2 \tag{12}$$

In the formula, $v$ is the kinematic viscosity, (m$^2$/s); slag, steel, l, 2 represent mold powder, molten steel, simulated mold powder and simulated molten steel, respectively. According to the various parameters in Table 5, the kinematic viscosity of water at room temperature is not much different from that of liquid steel. Using the mixture model of aviation kerosene and vacuum pump oil can ensure that the kinematic viscosity is similar to that of mold powder. The viscosity of the mixed oil selected in this experiment is 75 mm$^2$/s, the mixture ratio between 3# aviation kerosene and N100 vacuum pump oil is 0.11 L:2.46 L.

**Table 5.** Physical parameters of different substances [11].

| Liquid Phase | Density (kg/m$^3$) | Dynamic Viscosity (Pa·s) | Kinematic Viscosity (m$^2$/s) |
|---|---|---|---|
| Water | 1000 | $1.0 \times 10^{-3}$ | $1.0 \times 10^{-6}$ |
| Molten steel | 7020 | $6.5 \times 10^{-3}$ | $0.95 \times 10^{-6}$ |
| 3# aviation kerosene | 800 | $2.0 \times 10^{-3}$ | $2.5 \times 10^{-6}$ |
| N100 vacuum pump oil | 880 | $88.0 \times 10^{-3}$ | $100 \times 10^{-6}$ |
| Mold powder | 2500~2900 | $20{\sim}800 \times 10^{-3}$ | $8{\sim}296.3 \times 10^{-6}$ |

The experiment with the water model to physically verify the numerical simulation results employs two kinds of SEN in different eccentric distance conditions to observe the fluctuation of the liquid surface and the change of the flow field in the mold.

## 3. Experimental Results and Analysis

### 3.1. Effect of SEN Eccentric Distance on the Flow Field

Much research indicates that the drift phenomenon in industrial fields and production procedures is mainly reflected in the level fluctuation of the mold and the melting distribution of mold powder on the steel-slag interface. Mold level fluctuations can be measured by technical means, but the distribution status of mold fluxes cannot be quantitatively described. To describe and analyze the drift phenomenon of the mold quantitatively, the drift index (B) has been adopted in this experiment to measure eccentricity of the flow field [14], calculated by the method shown in Equation (13):

$$B = (F_R - F_L)/[(F_R + F_L)/2] \tag{13}$$

where $F_R$ and $F_L$ are the values of liquid surface fluctuation close to the wall area on two sides of the mold along the eccentric direction of SEN, which are defined by the data of wave height variation at the same position and recorded by the wave height sensor arranged near the wall on both sides of the SEN. The data in Equation (13) are the average wave heights over a period of time.

The eccentric distance of the SEN is the most significant factor affecting the drift index in a mold. The numerical simulation is applied to study and analyze the drift degree in the mold under the conditions of different eccentric distances of the original SEN. According to the preliminary simulation test, the casting speed of 0.3 m/min and the immersion depth of 160 mm are selected as the experimental conditions.

The distribution of the flow field of molten steel in the mold in which the original SEN is located in the middle position of the mold, which is in a non-eccentric state with an eccentric distance of 0 mm, is shown in Figure 4.

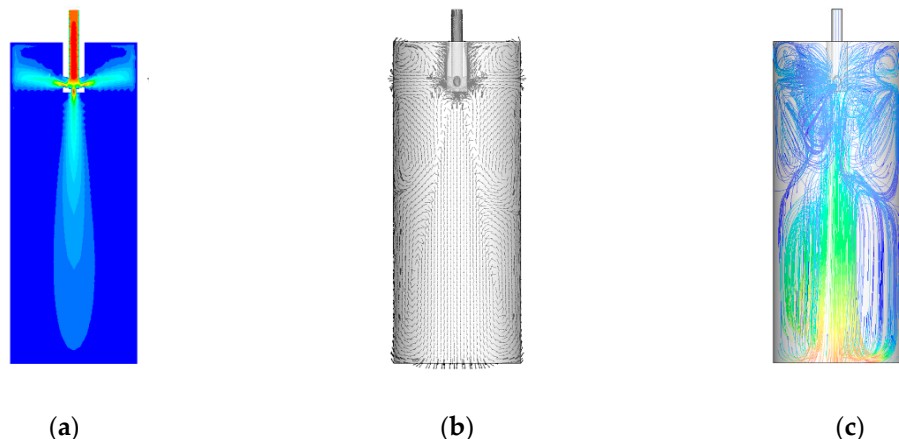

(**a**)　　　　　　　　　　(**b**)　　　　　　　　　　(**c**)

**Figure 4.** The distribution of flow field of molten steel in mold. (**a**) Contour; (**b**) Vector graph; (**c**) Streamline graph.

As shown in Figure 4, the molten steel through the bottom hole of SEN forms a vertical downward flow that is called a central stream. Its velocity gradually decreases with the increase of the impact depth, the shortest distance from the liquid surface of the mold to a certain position, in which the average downward velocity of the fluid on the cross section of the mold is equal to the casting speed, is defined as the impact depth. When the center stream reaches a certain depth, it forms an upward reflux stream. Similarly, the molten steel through the 4 side holes of the SEN forms 4 jets. The jets gradually expand and diverge in the flow process and their velocities are gradually reduced. When reaching and impacting the wall of mold, the jets can form upward and downward flows. The downward flows of molten steel flow to the center of the mold after the flows impact to a certain depth, forming a larger range of lower reflux area. Additionally, the downward flows are related to the distribution of flow field at the lower region of mold and the crystal structure after the round bloom enter the secondary cooling zone. The upward flows can disturb the steel-slag interface, but they have a great influence on the floating of the inclusions and the fluctuation of the liquid surface and the heating of the mold flux, and they determine the heat transfer of liquid surface and the fluctuating behavior of the steel-slag interface.

Figure 5 shows the velocity maps on the surface of YOZ (longitudinal section on the direction of casting speed) when the eccentric distances of the original SEN are 0, 12.5, 25, 37.5 and 50 mm in the casting condition. Among them, the cloud images represent the size of the molten steel flow field.

As shown in Figure 5, when the original SEN is in the eccentric condition and when the eccentric distance of the SEN increases, the center stream appears obviously asymmetrical, shifting to the edge of the round bloom, and it washes into the region of initial solidified shell. From the vector diagram of flow fields in Figure 5, it can be found that with the increase of the eccentric distance of the nozzle, the flow pattern of double reflux formed by the center stream near the outlet of the mold is gradually replaced by one large and one weak return flow, and eventually become only one large reflux stream, which is because the reflux stream in the eccentric direction of the nozzle is inhibited greatly. The continuous development of this one-sided large reflux stream leads to the deviation of the center stream. Meanwhile, the velocity of the free surface at the eccentric side is also largely imbalanced, up to 0.113 m/s when the eccentric distance of the SEN is 50 mm.

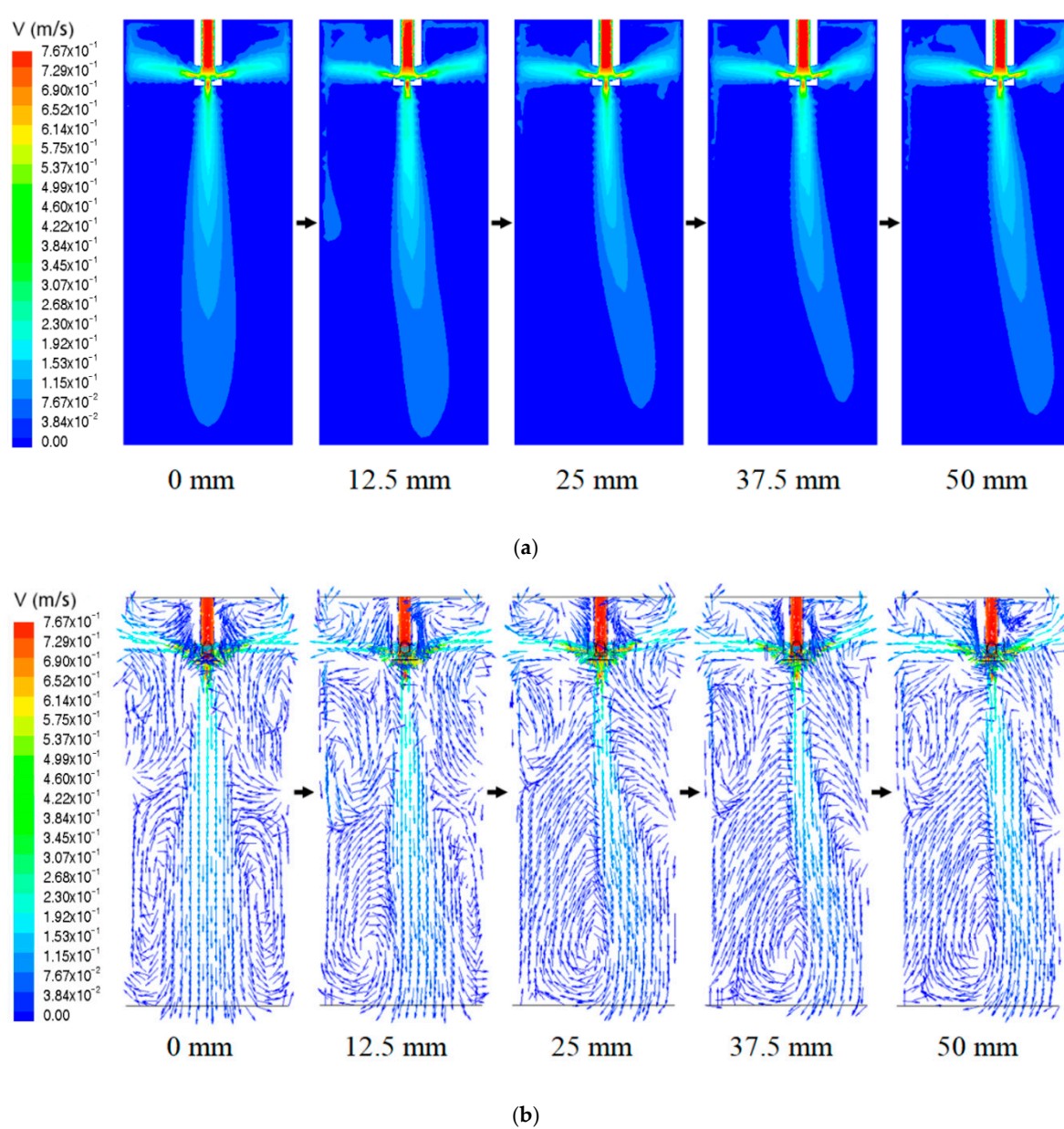

**Figure 5.** Flow fields of molten steel of the SEN with different eccentric distances. (**a**) Contour; (**b**) Vector graph.

The asymmetric flow field results in a significant difference in the impact degree on the shell from the side-hole jets around the nozzle. The stronger impact brings a higher-velocity upward stream and a greater disturbance to the meniscus. The difference of the impact degree leads to a great difference in the fluctuation of surface level on both sides of the nozzle. The large magnitude of surface level fluctuation at one side of mold can be propagated to the other side of mold with small magnitude of surface level fluctuation in the form of surface wave. The wave transfer process may cause a periodic fluctuation of liquid surface in the mold, which results in mold fluxes infiltration when the velocity of the free surface is too high. At the same time, the strong drift flow can greatly destroy the stability of the flow field and cause excessive fluctuations of the meniscus and an increase of the velocity, resulting in slag entrapment. This is called the phenomenon of drift flow or the drift phenomenon.

The velocity distribution of the steel-slag interface when the eccentric distance of the original SEN changes is shown in Figure 6, where it can be observed that when the SEN is located in the middle position of the mold, the flow velocities of both sides of the nozzle are basically the same in

the steel-slag interface. When the nozzle deviates from the mold center of 25 mm, there are obvious differences in velocities between the two sides of the nozzle in the steel-slag interface, increasing significantly on the deviated side of the SEN, which is due to the SEN deviation from the center of the mold. Additionally, the vortex center location of the backflow changes, causing upward reflux to be obviously strengthened, while the erosion for the initial solidified shell from molten steel flow on the deviated side of SEN also increases.

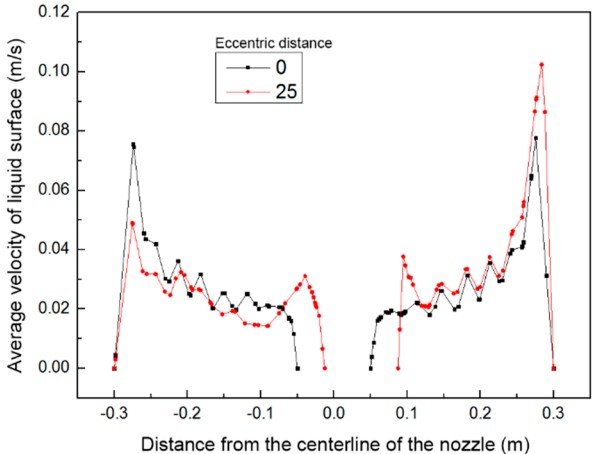

**Figure 6.** The effect of eccentric distance on the velocity of the steel-slag interface.

In order to research further the effect of drift phenomenon on bubble distribution and inclusion removal efficiency in the mold, a serious eccentric flow field with SEN eccentric distance of 50 mm was selected as an example to compare with the simulation results of the mold with SEN eccentric distance of 0 mm.

The distribution of argon bubbles in the mold is shown in Figure 7. As seen in Figure 7, when the nozzle is located in the center of the mold, the distribution of argon bubbles on both sides of the SEN is uniform, and the bubbles with diameter over 2 mm are mainly distributed around the nozzle. This is mainly due to large-size bubbles with larger buoyancy, which are less affected by the drag force of molten steel and can float quickly near the nozzle, but it is easy to cause the increase of fluctuation of liquid level near the nozzle. The bubbles with diameter over 1 mm are widely distributed in the flow field, which is mainly because the buoyancy of small-size bubbles is small, and the bubbles are greatly influenced by the drag force of molten steel, so that they can impact further with molten steel. When the eccentric distance of SEN is 50 mm, the symmetry of argon bubbles distribution on both sides of the SEN is less affected, and the main influence is that the distribution of argon bubbles in the whole mold is not uniform. On the side far from the eccentric direction of the nozzle, there is some areas that the argon bubbles cannot reach, which weakens the effect of argon blowing in the mold. The movement behavior of argon bubbles is similar to that of the non-eccentric SEN, but due to the effect of the deviation of the center stream, the disturbance of the flow field in the lower part of the mold increases, and the impact depth of most bubbles blown out from the bottom hole decreases. However, there are very few small bubbles separated from the main argon blowing range to obtain a larger impact depth. Considering the force of small-size bubbles, these separated bubbles are not easy to float with other bubbles, and can eventually stay in the solidified shell.

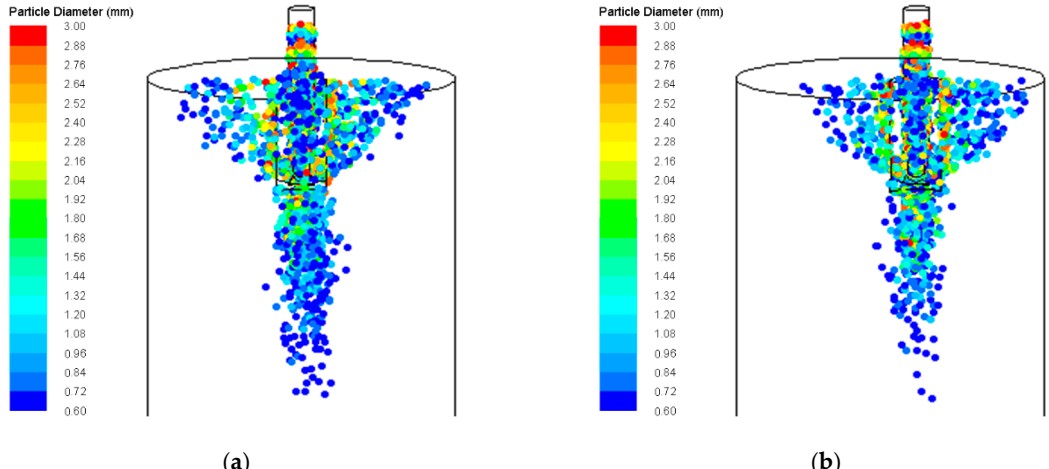

(**a**)                                                                                    (**b**)

**Figure 7.** Distribution of argon bubbles of different SEN eccentric distances. (**a**) SEN eccentric distances is 0 mm; (**b**) SEN eccentric distances is 50 mm.

The movement of inclusions in mold is also simulated in mathematical model. As shown in Figure 8, with the increase of eccentric distance of the SEN, the removal rate of inclusions in the mold decreases. The lowest removal rate is 40.10%, when the SEN eccentric distance is 50 mm, which decreases by more than 15.8% compared to the removal rate (47.64%) of the mold with the nozzle in the middle position. Because the small inclusions in the mold are mainly driven by the drag force of the molten steel, and in the eccentric flow field, the large drift flow of molten steel in the mold causes small inclusions to flow down with it and stay at the deeper location of the liquid core, but they do not float upward so that they are gathered inside the round bloom.

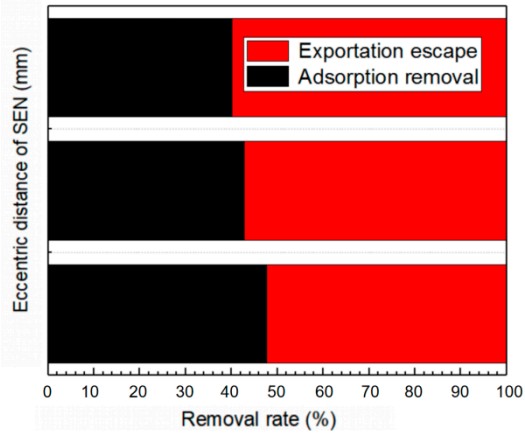

**Figure 8.** Inclusion removal rates of different SEN eccentric distances.

In order to study the effect of eccentric flow field on the temperature field of the mold and the solidification shell in the round bloom, the temperature distribution in different positions of the mold was studied by taking the SEN eccentric distance of 50 mm with larger drift phenomenon as an example, and compared with the temperature field of the mold with the SEN in the center position. Some representative positions were selected to analyze the temperature distribution, including the the radial temperature distribution of liquid level, radial temperature distribution of mold outlet plane (Z = 900 mm), the center axis temperature distribution of the mold and the center axis of the SEN. Because the nozzle deviates from the center axis of the mold, the temperature difference between the center axis of the mold and the center axis of the SEN is selected to judge the temperature distribution inside the mold.

Under the same conditions, the effects of the nozzle eccentric distances (0, 12.5, 25, 37.5 and 50 mm) for the drift index of the mold are shown in Figure 9. According to the data in Figure 9, the theoretical relationship between the eccentric distance data and the drift index (B) of mold is derived, as shown in Equation (14):

$$B = a \times [1 - \exp(-b \times D)] \tag{14}$$

where $D$ represents the values of the eccentric distance of SEN, mm; and a and b are the empirical constants, a = −0.021, b = −0.063.

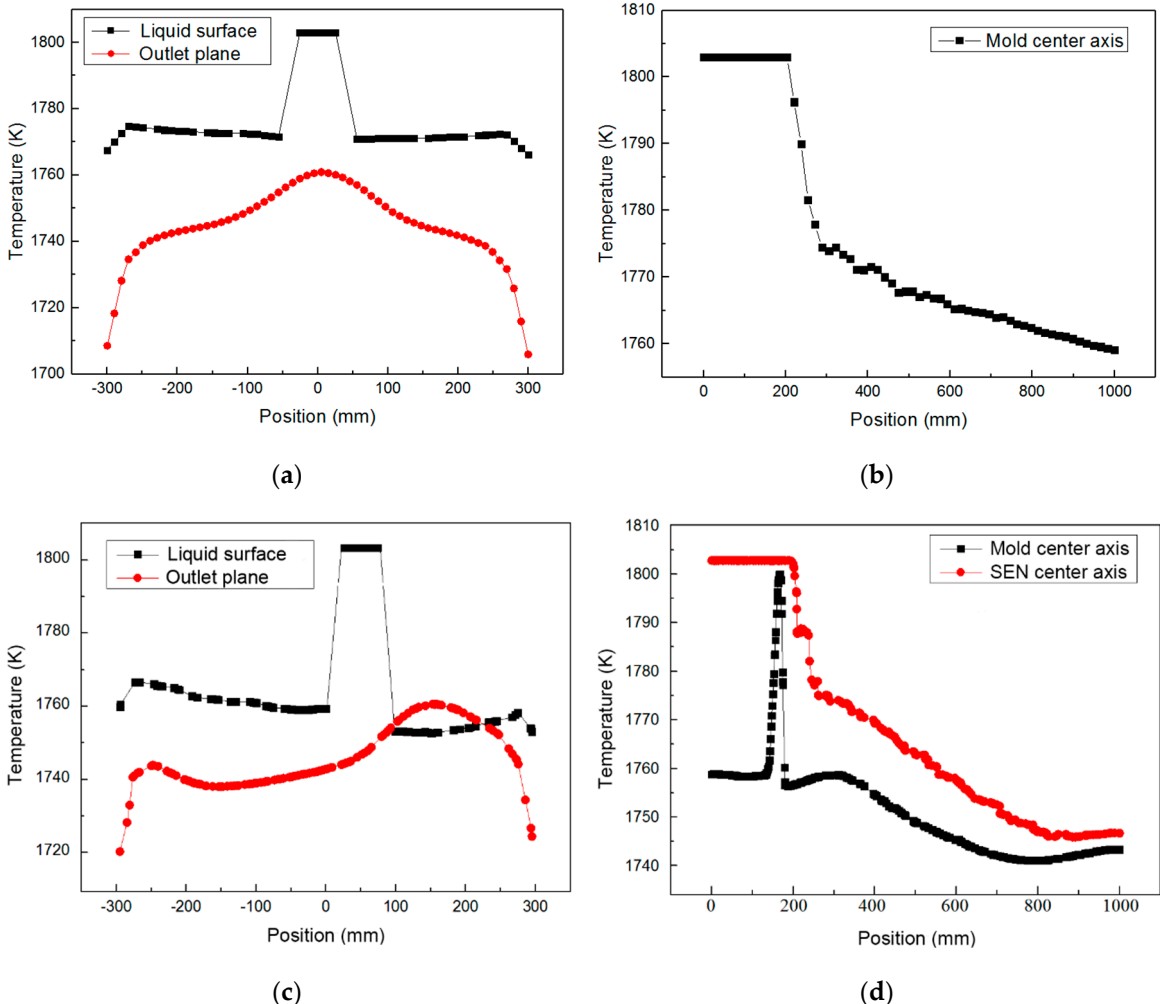

**Figure 9.** The effect of eccentric distance on the temperature field of mold when: (**a**) The temperature distribution along radial direction as the SEN eccentric distance is 0 mm; (**b**) The temperature distribution along central line as the SEN eccentric distance is 0 mm; (**c**) The temperature distribution along radial direction as the SEN eccentric distance is 50 mm; (**d**) The temperature distribution along.

From Figure 10 and Equation (14), it is concluded that there exists an exponential relationship between the drift index (*B*) and the eccentric distance (*D*) of SEN, and the drift index of mold increases with the increase of SEN eccentric distance. When the eccentric distance is less than 5% (30 mm) of the diameter of the mold, the drift index increases slightly, but both are less than 0.1. At this time, the peak value of the stream at the eccentric side of the nozzle is higher than the other side, yet the basic type of flow field also remains unchanged. When the eccentric distance of nozzle is more than 5% (30 mm) of the mold diameter; however, the drift index increases dramatically and the maximum is over 0.45, in which the velocity distribution of flow field in the mold completely changes and the disturbance of the liquid surface from the stream of the eccentric side increases. It is easy to form a vortex near the

nozzle of the mold, which can cause adverse effects such as slag entrapment. Therefore, when the eccentric distance of SEN is 50 mm, the phenomenon of eccentric flow is most serious. Based on the above analysis and the actual production situation, the eccentric distance of 50 mm is selected to study eccentric flow field and optimize the SEN.

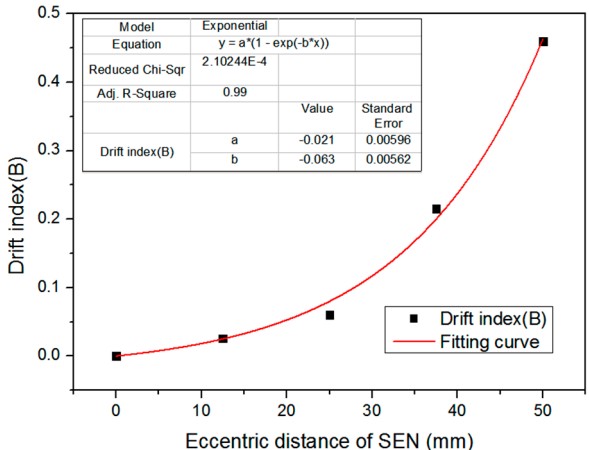

**Figure 10.** The effect of eccentric distance on the drift index.

### 3.2. Effect of Selection of the Bottom Structure of the SEN on Eccentric Flow Field

In the selection and optimization of the SEN, the existence of the bottom outlet has a great influence on the flow field in the mold. The original SEN is the nozzle with a bottom hole, so the influence of the bottom shapes of the SEN on the flow field of mold is directly investigated under eccentric casting conditions. When the casting speed is 0.3 m/min with an immersion depth of 160 mm and eccentric distance is 50 mm, the distribution of turbulent kinetic energy in the liquid surface of the mold, with and without a bottom hole, is shown in Figure 11.

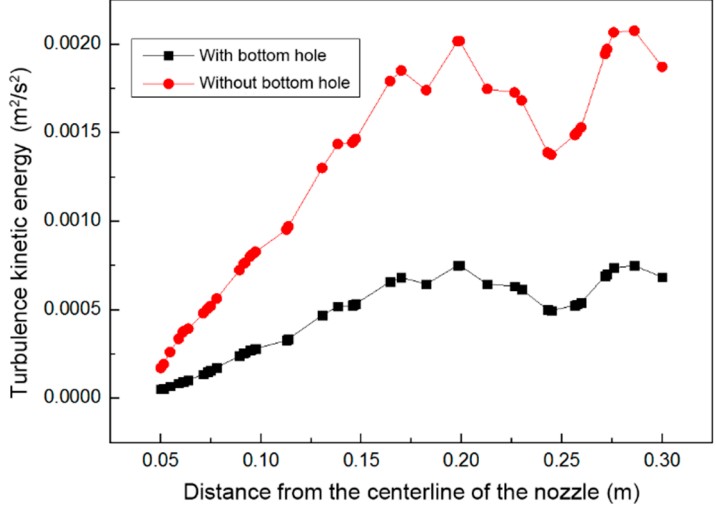

**Figure 11.** Turbulent kinetic energy distribution of liquid surface with different nozzle bottoms.

From Figure 11, it is seen that when the SEN has a bottom outlet, the turbulent kinetic energy in the mold decreases significantly. This is because the existence of the bottom outlet can play the role of distribution. The dip angle of the side hole in the SEN is upward-sloping to effectively improve the temperature of the steel-slag interface, and it is also beneficial to the removal of inclusions.

However, when the mold is at state of high casting speed, the upward-sloping nozzle can aggravate level fluctuation. The bottom outlet can split the molten steel flows of four side holes so that the

velocity of liquid steel inside the hole is not too large to cause slag entrapment or the liquid surface to be exposed.

Under the same conditions and at an eccentric distance of 50 mm, pressure distribution at the different SEN bottoms is shown in Figure 12. As the SEN bottom has an outlet, the pressure of the SEN bottom from molten steel is greatly reduced, which is beneficial to decrease the impact of molten steel on the SEN and therefore is of positive significance in improving the service life of the SEN.

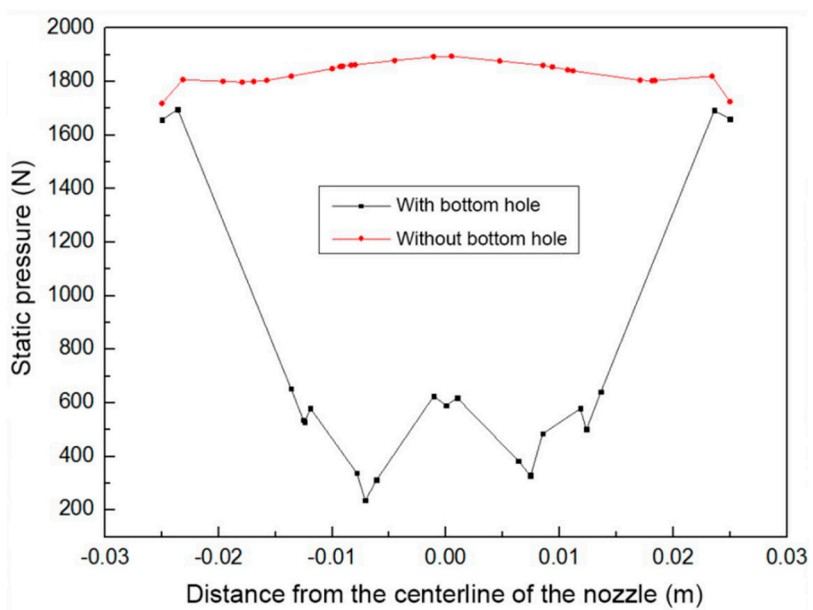

**Figure 12.** Variation of pressure at different bottoms of a submerged nozzle.

The above analysis shows that, under the eccentric casting condition, the SEN with a bottom outlet can significantly reduce the bottom pressure and turbulent kinetic energy, weaken the erosion of the nozzle and decrease the fluctuation of the liquid surface. Therefore, the bottom outlet plays an important role in the eccentric flow field improvement.

*3.3. Effect of Selection of the Rotation Angle of the Side Hole on Eccentric Flow Field*

3.3.1. Effect of the Rotation Angle of the Side Hole on Flow Field

The choice of different rotation angles on the side hole of the SEN can affect the flow field distribution of the mold in different degrees. According to the analysis of Section 3.2, based on reservation of the bottom outlet, the effects of rotation angles (0°, 15° and 30°) on liquid wave height, turbulent kinetic energy and impact depth of the mold are analyzed in the following section.

Figure 13 shows the velocity vector diagram of the steel-slag interface. Combined with Figure 4, it can be found that under the condition of non-eccentric flow field, the four upward-sloping side holes of the SEN bring four upward streams to the liquid surface of mold. The streams then hit the wall of the mold, and a part of the molten steel continues to rise, flowing through the steel-slag interface and reflowing down near the nozzle, resulting in four high-speed regions and the vortex core regions in a direction perpendicular to the liquid surface, which can bring enough heat fluxes and flows. However, excessively high speed can also cause adverse effects such as slag entrapment.

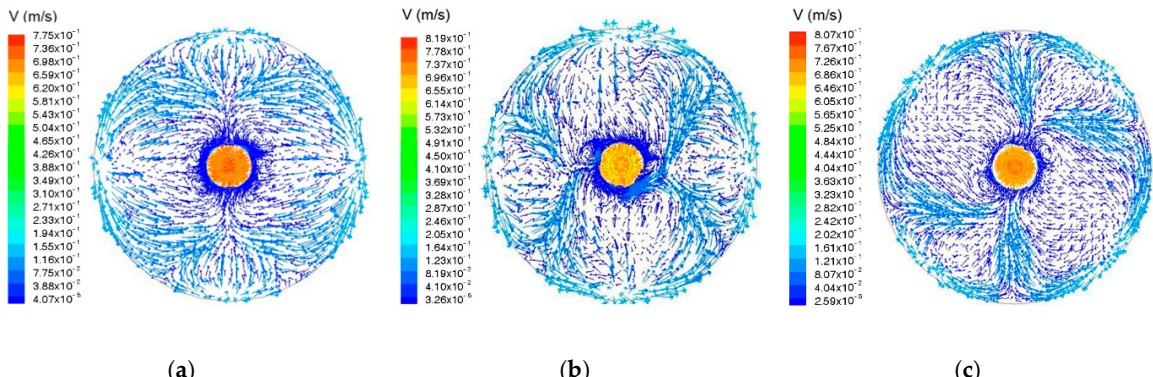

**Figure 13.** Velocity vector of the steel-slag interface in different rotation angles of the side hole. (**a**) The rotation angles on the side hole of the SEN is 0°; (**b**) The rotation angles on the side hole of the SEN is 15°; (**c**) The rotation angles on the side hole of the SEN is 30°.

In Figure 13, when the rotation angle of the side hole is not 0°, four upward flows impact the wall of mold along the vertical direction, rise to the steel slag interface, and then reflow evenly to the nozzle, and the liquid surface is stable. However, the molten steel that directly reflows from the nozzle has a large downward velocity, increasing the impact depth of the center stream. When the rotation angle of side hole is 15°, four rising streams impact the wall of mold according to a certain angle of rotation. Thus, the molten steel that reaches the steel-slag interface also forms four high speed streams with a certain angle of rotation. Additionally, the horizontal swirling flow around the nozzle is formed to weaken the downward velocity of the reflux, but the process that the molten steel reflows evenly to the nozzle is still partially present. When the rotation angle of the side hole is 30°, the process of direct even reflux in the steel-slag interface is completely replaced by the swirling process of four high speed streams, forming a plurality of horizontal swirling centers near the nozzle, a phenomenon of swirling flow that is more obvious.

When the SEN eccentric distance is 0 mm with a casting speed of 0.3 m/min and an SEN immersion depth of 160 mm, different rotation angles of the side holes cause effects on the liquid wave of the mold and the turbulence kinetic energy of liquid surface, as shown in Figure 14.

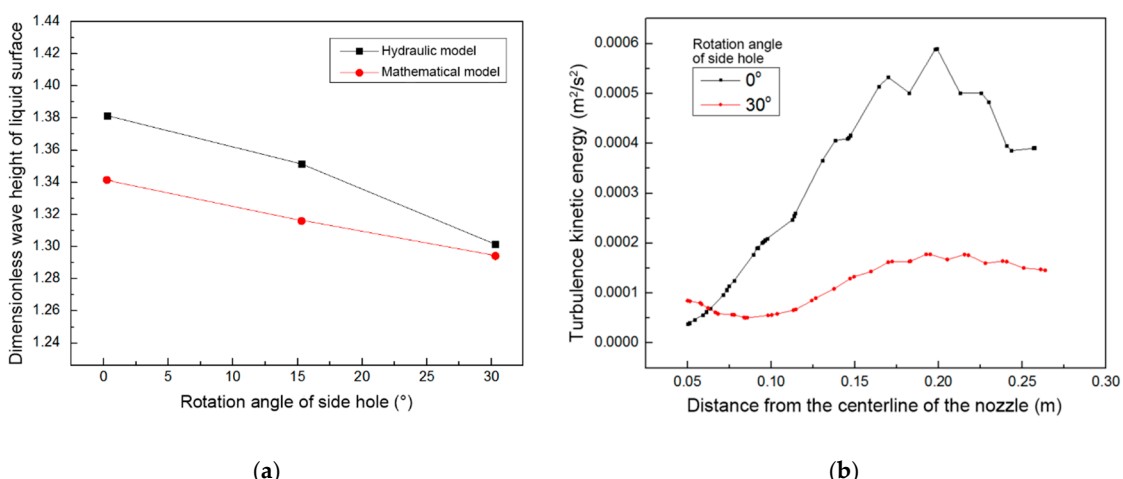

**Figure 14.** The effect of rotation angle of side holes to liquid surface of the mold. (**a**) Liquid wave height; (**b**) Turbulence kinetic energy of liquid surface.

In order to make the measured wave height data of numerical simulation and modeling experiment comparable, all wave height data have become dimensionless:

$$F_D = 100 F_S / (\lambda D) \tag{15}$$

In this formula, $F_D$ represents dimensionless wave height; $F_S$ is the measured wave height data of liquid surface, mm; $\lambda$ and $D$ represent the value of similarity ratio and the horizontal section diameter of different mold models, mm.

Figure 14 shows that when the outlets of side holes have rotation angles at tangential directions, the average wave height of the liquid surface tends to decrease with increasing rotation angle. The side holes with rotation angle can obviously reduce the turbulent kinetic energy of the mold surface and eliminate the peak region of turbulent kinetic energy. This is because the rotation angle of the side holes increases the horizontal velocity of streams and brings the horizontal swirls to the mold when the molten steel flows out through SEN. The existence of horizontal swirling flows can weaken the velocity of the molten steel flow in the Z direction (axis of mold) so that the upward reflux velocity decreases, thereby reducing the vortex center position of upward reflux and disturbance to the free surface from it. In this condition, the liquid wave height of the mold also decreases. At the same time, the elimination of the peak value in Figure 14b can effectively prevent the excessive fluctuation of the area that can cause many problems, such as liquid steel exposure.

The side-hole jets and central stream have comprehensively effect on impact depth. As shown in Figure 15, with the increase of the rotation angle of the side hole, the impact depth of the stream in the mold gradually decreases. Like in Figure 14, this is because of the rotation angle so that swirling flows in the horizontal direction are formed in the mold. The presence of these swirling flows can disperse the initial kinetic energy of molten steel and reduce the velocity component in the Z direction; the horizontal swirls also improve the impact so that the streams are gentler. Therefore, the increase of the rotation angle can decrease the impact depth. The greater the rotation angle and the higher the intensity of horizontal swirling flows are, the more obvious the effect will be. From the above analysis, it is concluded that the horizontal swirling flows occur in the mold with the increase of the rotation angle of the side holes with other conditions unchanged. The impact depth of stream and the fluctuation of the liquid surface gradually reduce, and the turbulent kinetic energy of liquid surface decreases significantly. To fully utilize the metallurgical capability of horizontal swirling flows, it is recommended that the rotation angle of the side hole be 30°.

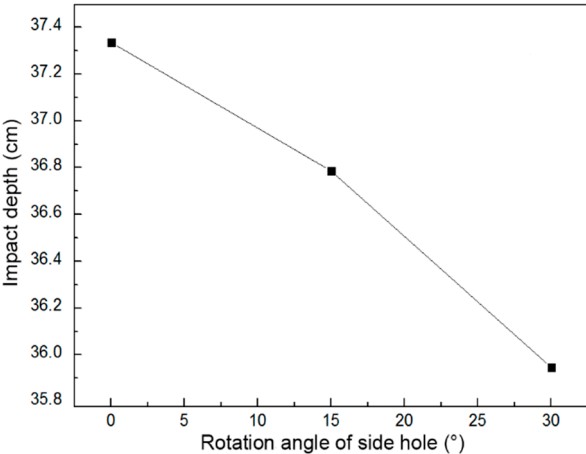

**Figure 15.** The effect of rotation angle of side holes for the impact depth of liquid steel.

3.3.2. Effect of the Swirling SEN Under Eccentric Casting

Under the condition of eccentric casting, the flow field behaviors of the mold employing the optimized swirling SEN, which is the swirling SEN with the rotation angle of side holes of 30°, are analyzed to verify the metallurgical effect of the optimized parameters in the following section.

Figure 16 shows the distribution of flow fields of the optimized swirling SEN in mold when the casting speed is 0.3 m/min with an immersion depth of 160 mm and an eccentric distance of 0 or 50 mm. Comparing the flow fields with those of the original nozzle (Figure 5), Figure 16 indicates that, when using the swirling SEN, the side holes with a tangential rotation angle can form horizontal swirling flows in the mold, weakening the impact of high speed jets on the wall of the mold. The ability of swirling SEN to form a wide range of horizontal swirling flow inside the mold is similar to that of the M-EMS (Danieli Metallurgical Equipment & Service (China) Co Ltd, Changshu, China) device, which accelerates the horizontal movement of molten steel by electromagnetic force and forms swirling flow field inside the mold. The maximum horizontal swirling velocity of molten steel near the outlet of the side hole of the swirling SEN can reach 0.6 m/s, and the maximum swirling velocity near the wall of mold is over 0.075 m/s, which is close to the maximum horizontal swirling velocity (generally 0.05~0.3 m/s) near the wall of mold when using the M-EMS device [15–18], so it can be considered that the swirling SEN has a similar metallurgical effect with M-EMS to a certain extent. The horizontal swirling flows formed by swirling nozzle is beneficial to more molten steel moving towards the shell. And to a certain extent, its tangential velocity promotes the breakage of the front end of columnar crystals by molten steel to form a large number of dendrite fragments, which can be used as crystal nuclei, and increase the probability of equiaxed crystal formation. Moreover, the horizontal swirls at the meniscus can stabilize the meniscus fluctuation, reduce the probability of slag entrapment, and provide a heat source for the meniscus to improve molten slag. The molten steel flowing out from the side hole also forms double streams of the rising and falling flows. Compared with the original SEN, however, the existence of horizontal swirling flows weakens the velocity of double streams in the Z direction and the reflux center has been greatly improved. Based on utilization of the optimized swirling nozzle in an eccentric distance of 50 mm, the bias phenomenon on center stream has been effectively restrained without noticeable eccentric stream.

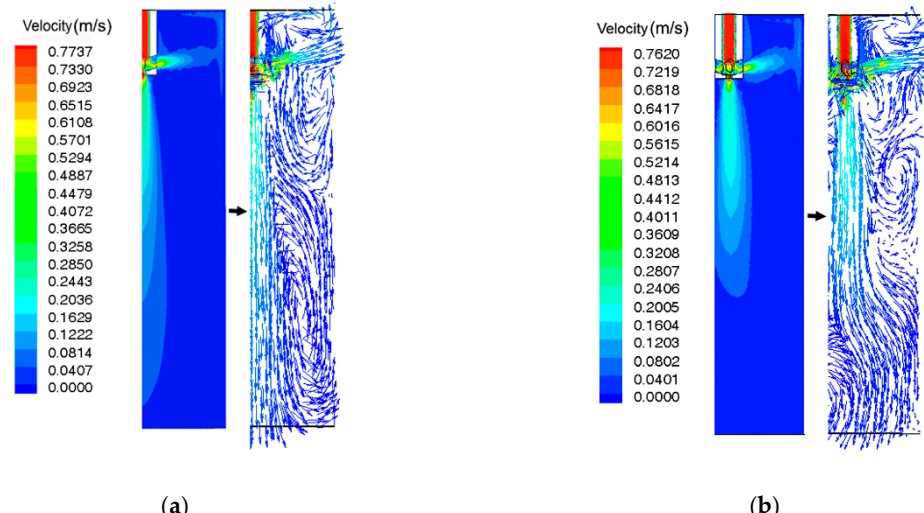

(**a**)　　　　　　　　　　　　　　　　(**b**)

**Figure 16.** Flow fields of molten steel in the mold with different conditions. (**a**) Five-furcated SEN with outlets in tangential direction with the same axis; (**b**) Five-furcated SEN with outlets in tangential direction with deviation from the center of 50 mm.

Figure 17 shows the velocity vector diagram of the steel-slag interface with the original nozzle and the optimized swirling nozzle under the eccentric condition. As seen in Figure 17, compared

with the non-eccentric condition in Figure 13, when the nozzle eccentric distance is 50 mm, using the original nozzle, the molten steel close to the eccentric sidewall of the mold can still return uniformly to the nozzle. However, at the same time, the liquid steel of other areas forms an auxiliary reflux center near the center area of mold, which is the single high-speed reflux vortex and can accelerate the reflux process for molten steel far away from the region of the nozzle to compensate for the structural unbalance of the flow field caused by the nozzle that deviates from the center. The single high-speed reflux vortex can, however, easily cause slag entrapment of molten steel. After using the optimized swirling nozzle, compared with the single reflux vortex in the flow field of the original nozzle, the swirling nozzle produces four high-speed upward streams with the same rotation angle so that the widespread horizontal swirling flows are formed in the flow field of mold. Therefore, there are also multiple centers of horizontal swirling flows formed close to the wall of the mold in the steel-slag interface. The maximum horizontal swirling velocity is 0.045 m/s, and the downward velocity in the center of swirling flows is less than 0.015 m/s, which is much lower than the minimum critical velocity for slag entrainment of 0.264 m/s [19]. Furthermore, the density of the vector in the velocity vector diagram of the swirling SEN decreases greatly compared with the diagram of the ordinary SEN under the same conditions, this indicates that the high-speed flows reduce significantly and the steel-slag interface is more stable. Accordingly, these not only greatly weaken the size and velocity of the reflux vortex near the center of the interface to suppress the phenomenon of slag entrapment but also stabilize the flow field of molten steel, effectively reducing the inconsistency of reflux process in different regions. At the same time, because the optimized swirling SEN can form similar horizontal swirling flows around the center stream at the lower region of mold, it can also greatly inhibit the bias phenomenon of center stream in the mold.

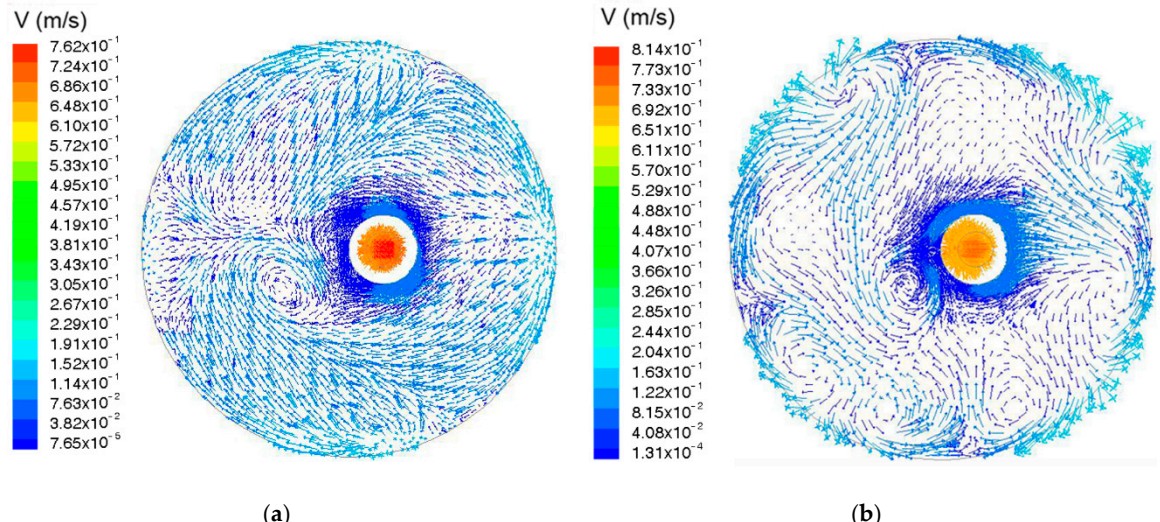

(**a**)                                          (**b**)

**Figure 17.** Velocity vector of steel-slag interface under the eccentric condition. (**a**) The original SEN; (**b**) The optimized swirling SEN.

Compared with Figures 18a and 8, the optimized swirling SEN can improve the overall removal rate of 1~50 μm inclusions, and the maximum removal rate is 51.70%. However, under the SEN eccentric conditions, the overall removal efficiency of inclusions in the mold still decreased from 51.7% to 50.36%, and the difference is only 2.6%. But compared with the removal rate under different conditions of the original nozzle, the removal efficiency is still improved.

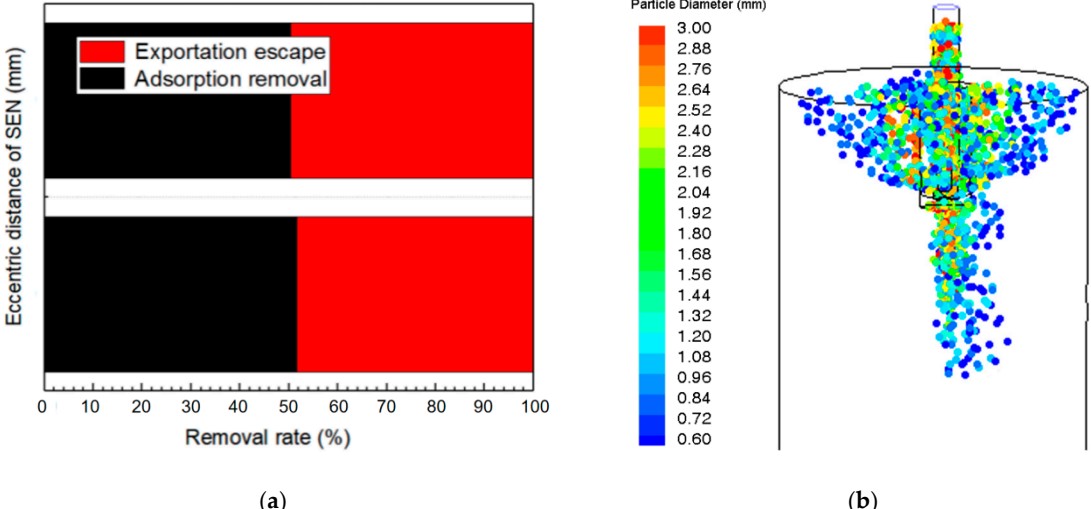

**Figure 18.** Inclusion removal rates and distribution of bubbles for the optimized swirling SEN. (**a**) Inclusion removal rates; (**b**) Distribution of argon bubbles.

As shown in Figure 18b, under the SEN eccentric conditions, compared with the distribution of bubbles in the original SEN of Figure 7, when argon is blown by optimized swirling SEN, the horizontal swirling flow can effectively promote the horizontal movement of bubbles, inhibit the non-uniformity of bubbles distribution in the mold caused by bias flow, reduce the impact depth of bubbles, and at the same time, make the large-size bubbles with a diameter of more than 2 mm move farther in the horizontal direction so that the liquid level fluctuation near the SEN is weakened.

By comparing the temperature distributions of different positions in the mold shown in Figures 9 and 19, when using the optimized swirling SEN, the temperature difference between the center axis of the mold and the center axis of the SEN has been eliminated below 400 mm. Although there is still a certain temperature difference on the liquid surface of the mold, it is controlled within 5 K. The temperature distribution at the outlet plane (Z = 900 mm) of the mold is uniform and the temperature in most areas of the center is controlled around 1750 K. Compared with the original SEN, the temperature at the outlet plane (over 1760 K) decreases by more than 10 K which can promote the uniform distribution of the shell thickness around the cross section of the mold. All these indicate that the eccentric phenomenon of the temperature field distribution caused by the drift flow of molten steel has been greatly improved.

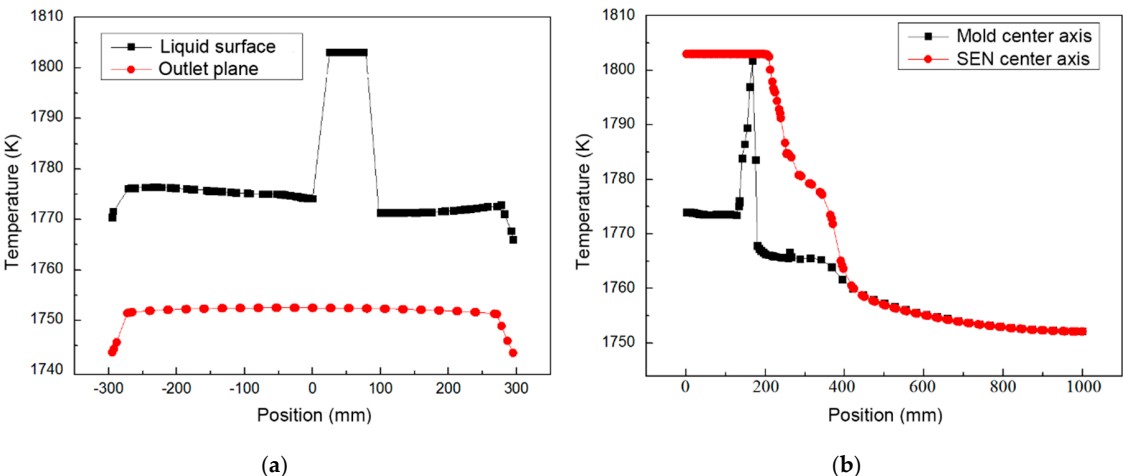

**Figure 19.** Temperature field of the mold with the optimized swirling SEN. (**a**) The temperature distribution along radial direction; (**b**) The temperature distribution along the center line.

As shown in Figure 20, in the eccentric flow field, the optimized swirling nozzle has a shallower impact depth than the original nozzle, which is beneficial to floating and removing inclusions. When the eccentric distance reaches 50 mm, the drift index of the original SEN reaches 0.46, which causes the flow field to be completely unbalanced with turbulent kinetic energy and the surface disturbance increases. However, in the eccentric casting condition, the optimized swirling nozzle can obviously reduce the drift index to 0.28, a decline of more than 39%. In such conditions the drift phenomenon of the mold is suppressed in the lower level so that it does not affect the normal distribution of flow field.

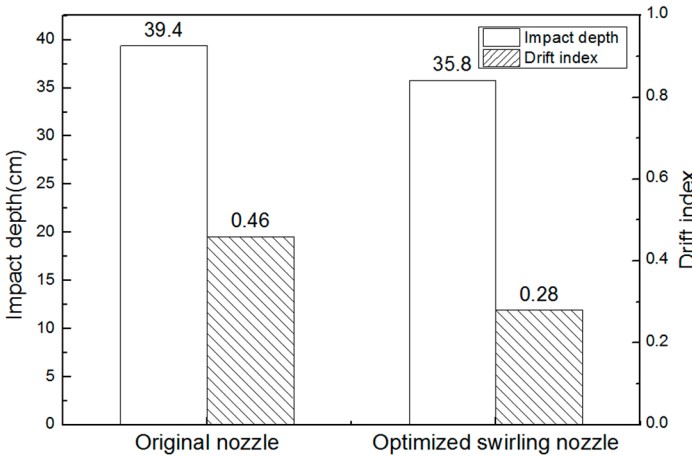

**Figure 20.** Impact depth and drift index of molten steel for SEN before and after optimization.

*3.4. Comparison with Water Modeling Experiment*

According to the simulation results of different SEN structures and process parameters obtained in the previous experiments, the structural parameters of SEN have been optimized and the optimized swirling nozzle is as follows: five-furcated nozzle with a bottom hole and four side holes, the tangential rotation angle of side holes is 30°. Under the same parameters of mold, the water modeling experiment has been applied as a contrast test to verify the practical effect of the optimized swirling nozzle and the original nozzle in the eccentric flow field.

As shown in Figure 21, according to the results of experiment in the eccentric distance of 50 mm, it is found that the flow field in the mold employing the original nozzle is obviously biased to the side on which the nozzle is close to the wall. Two seconds after tracer injection, there is an obvious imbalance in the flow field of mold. In contrast, with the use of optimized swirling nozzle casting, the central stream of molten steel does not give rise to bias flows and the nonuniform flow field has been significantly improved. The results of water modeling experiment and numerical simulation in the previous sections are consistent.

Figure 22 shows that with the use of the original nozzle in the process of eccentric casting, the phenomenon of slag entrapment is obviously observed (Figure 22a). In addition, when the casting speed increases to 0.4 m/min, the mold powder at the stream impact point is washed away by high-speed liquid flow so that it leads to surface exposure of liquid steel. However, after using the optimized swirling nozzle in Figure 22b, there is no slag entrapment or molten steel exposed in the mold under the same conditions of casting speed. This illustrates that the optimized swirling nozzle not only can effectively suppress the drift flow but also can significantly reduce the quality defects of the round bloom.

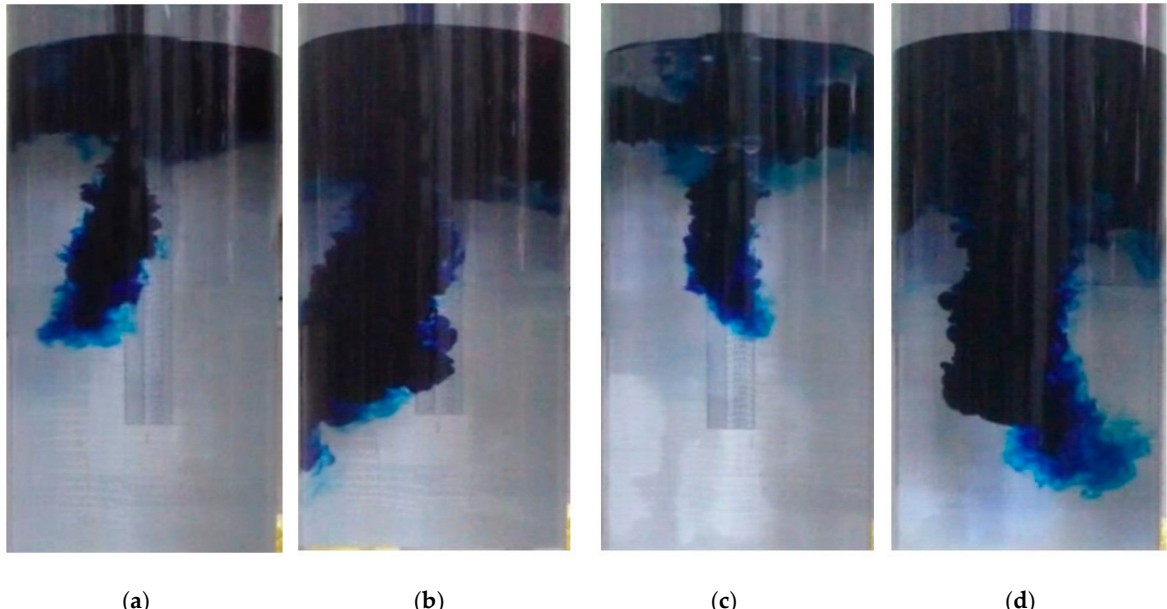

|         |         |         |         |
|:-------:|:-------:|:-------:|:-------:|
| (**a**) | (**b**) | (**c**) | (**d**) |

**Figure 21.** The flow field of the mold at different times while using different SENs in water model. (**a**) Original nozzle at $t = 0.5$ s; (**b**) Original nozzle at $t = 2$ s; (**c**) Optimized swirling nozzle at $t = 0.5$ s; (**d**) Optimized swirling nozzle at $t = 2$ s.

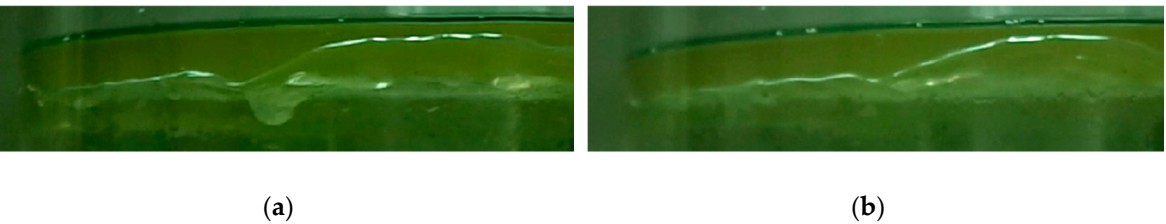

|                 |                 |
|:---------------:|:---------------:|
|     (**a**)     |     (**b**)     |

**Figure 22.** Morphology of steel-slag interface in mold for SEN before and after optimization. (**a**) Original nozzle; (**b**) Optimized swirling nozzle.

## 4. Conclusions

In this study, the flow fields of a mold equipped with different SEN have been simulated through the method of physical and mathematical models. The effects of SEN structure parameters for turbulent kinetic energy, the impact depth and drift index are analyzed and optimized in the mold when the SEN is in an eccentric state, namely, a mold in an eccentric flow field. The following aspects are concluded in this paper:

(1) The eccentric SEN can result in serious bias flow in the mold, causing asymmetrical distribution of flow field for molten steel. The drift index reaches up to 0.46 in the eccentric distance of 50 mm, which leads to problems of slag entrapment and liquid steel exposed in production.

(2) With the increase of the rotation angle on the SEN side holes, the horizontal swirling flows are formed in the flow field of mold so that the impact depth of the stream and turbulent kinetic energy of liquid surface decrease accordingly.

(3) The SEN with a bottom outlet can significantly reduce the bottom pressure and turbulent kinetic energy and weaken the scour of level fluctuation and molten steel for the SEN, thereby prolonging the service life of nozzle.

(4) In the eccentric casting condition with an eccentric distance of 50 mm, the use of the optimized swirling SEN, which employed a five-furcated nozzle with a bottom hole and four side holes and a tangential rotation angle of the side holes of 30°, can effectively inhibit the phenomenon of bias flow in the mold and reduce the mold drift index to 0.28, a decline of more than 39%. Compared

to a water modeling experiment, it is found that the optimized swirling nozzle can greatly reduce eccentric flow field and the level fluctuation significantly so that slag entrapment and exposed molten steel disappear.

(5)　The optimized swirling SEN can improve the inclusion removal efficiency in model although it is used in the eccentric casting condition, so it is suggested that the optimized swirling SEN should be used in actual continuous casting operation with eccentric casting condition.

(6)　The conclusions of the mathematical and physical models in this paper are related to similarity in non-perfect conditions and must be further compared to industrial experience which can be made in future.

**Author Contributions:** All authors contributed significantly. Y.J., C.C. and Y.L. designed the project. P.L., Y.J., F.Y. and Z.L. performed the numerical simulations and data collection. P.L., Y.J., F.Y. and Z.L. analyzed the data. All authors contributed to the discussion of the results. P.L. and Y.J. wrote and revised the manuscript. All authors have read and agreed to the published version of the manuscript.

**Funding:** This research was funded by the National Nature Science Foundation of China (NSFC) under Grant No. 51974213 and 51874215.

**Conflicts of Interest:** The authors declare no conflict of interest.

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
