# Peer review of "A Simulation and Optimization Study of the Swirling Nozzle for Eccentric Flow Fields of Round Molds"

_metals, doi:10.3390/met10050691_

Round 1

Reviewer 1 Report

Although it can be considered a bit long, the article is complex and argued by the results of research. Indeed, in the continuous casting, nozzle position may deviate from the center under the operating conditions, which may cause periodic fluctuation of the steel-slag interface. Therefore, it is very important for the control of the flow field  in the mold. The authors manage to encompass all aspects of the problem, using the method of similarity, although the method is valuable for operating in non-perfect conditions (water modeling experiment). It will be interesting to see and compare it with the practical industrial values, from the steel continuous casting.

Author Response

Point 1: The authors manage to encompass all aspects of the problem, using the method of similarity, although the method is valuable for operating in non-perfect conditions (water modeling experiment). It will be interesting to see and compare it with the practical industrial values, from the steel continuous casting. 

Response 1: As the experiment in the round mould of an actual continuous caster in a steel mill is still in schedule, the practical industrial values of our study will be opened in the future. Sorry Sir!

Reviewer 2 Report

The presented paper “Simulation and Optimization Study of the Swirling Nozzle for Eccentric Flow Fields of Round Molds” depicts fluid flow of the molten steel in the round mould. Experimental works, including development of the mould water model, along with the numerical modelling have been presented by the Authors. The research methodology presented in the paper is very clear and only minor revision has to be done.

  • Please check again formulas in the manuscript and the symbols description (for example in the formula 8 there is "da" but in the description dwith index - line 118).
  • Heat transfer coefficient unit is W/m2K ( the third type of the boundary condition) – please see table 3. It has to be addressed.
  • All abbreviations used in the text should be explained – with the first occurrence
  • Based on the presented results, for sure Authors are able to provide more conclusions supporting industrial application. What are the best working parameters for the SEN in the round mould?
  • The correct thickness of the shell solidifying within the primary cooling zone of the continuous casting machine is amongst the most important parameters assuring the safe operation of the casting process. What is the impact of the position of the SEN on the shell in the mould?
  • Position of the SEN - during casting - is changing. Have the Authors analyzed different immersion depth of the SEN versus distance from the center?

Author Response

Thank you for your kind comments, they are really very helpful.

Point 1: Please check again formulas in the manuscript and the symbols description (for example in the formula 8 there is "da" but in the description da with index - line 118). 

Response 1: We check formulas, and  da in the formula 8 and the description -line 118 are all replaced to da,avg. Thanks a lot!

Point 2: Heat transfer coefficient unit is W/m2K ( the third type of the boundary condition) – please see table 3. It has to be addressed.

Response 2: We correct the Heat capacity unit from J/kg to J/(kg·K), replace the transfer coefficienct  to effective thermal conductivity according to equation (5), and the unit of effective thermal conductivity is corrected to W/(m·K) in table 3. The data used in table 3 is addressed from reference article[12], which is added in line152.

Point 3: All abbreviations used in the text should be explained – with the first occurrence.

Response 3: The explain of SEN in abstract is added-line2.

Point 4: Based on the presented results, for sure Authors are able to provide more conclusions.

Response 4: Conclusion 5# is added in the article, which shows that optimized swirling nozzle can improve the inclusion removal effeciency-line 591-594.

Point 5: Supporting industrial application. What are the best working parameters for the SEN in the round mould?

Response 5: Conclusion 4# is shows that the working parameters of optimized swirling nozzle can improve the inclusion removal effeciency-line 584-590. And Conclusion 5# suggests the applicaion of the optimized swirling nozzle in actual caster-line 591-594.

Point 6: The correct thickness of the shell solidifying within the primary cooling zone of the continuous casting machine is amongst the most important parameters assuring the safe operation of the casting process. What is the impact of the position of the SEN on the shell in the mould?

Response 6: As the simulation of the solidification of shell in the mold is complex, and our article is long and complex now. We think it is better to establish an article to analyze the effect of the eccentric distance to the thickness of the shell.

Point 7: The correct thickness of the shell solidifying within the primary cooling zone of the continuous casting machine is amongst the most important parameters assuring the safe operation of the casting process. What is the impact of the position of the SEN on the shell in the mould?

Response 7: From Fig. 9 we can see that when the eccentric distance is reached the maximum(50mm) in our article, the temperature of the position near the wall of mould at the outlet level of SEN is decreased by less than 20K, as compared to that for zero eccentric SEN-lines 319-323. We think this difference of temperature will not cause safty issues, but will cause serious quality issues. As the simulation of the solidification of shell in the mold is complex, and our article is long and complex now. We think it is better to establish an article to analyze the effect of the eccentric distance to the thickness of the shell. Thanks very much.

Point 8: Position of the SEN - during casting - is changing. Have the Authors analyzed different immersion depth of the SEN versus distance from the center?

Response 8: We think this issue is very important. However, our article is too long to add new analysis. We think it is better to discuss different immersion depth of the SEN versus distance from the center in next article. Thanks again.

Reviewer 3 Report

In the paper, the flow from submerged entry nozzle (SEN) into the mould is clearly elaborated by the method of numerical simulations.

The article lacks a more detailed literary search focused on the effects of the quantities that are analyzed in the article, on the real casting process and quality.

From the beginning, it is not explicitly stated whether the model calculates coupled flowing and cooling. The reader learns about it gradually and the information is inaccurate and incomplete. Thermal boundary conditions are not exactly described, even the most important boundary condition of heat flux into the mould wall.

The physical model is only isothermal. Either the two models cannot be compared, or it must be proved that the effect of steel cooling on the flow is negligible.

Although the results are interesting, they largely prove what is already known or what can be expected. The article has mainly theoretical or academic value and demonstrates mastering of the use of commercial software. The real conditions are much more complicated. On the steel level, the interface between the casting powder and the molten steel changes dynamically mainly due to the oscillation of the mould and due to the interventions of the steel level control system. The interface between the real solid shell and the liquid phase is not a smooth cylinder; the shell thickness is not uniform etc.

It is not clear, whether the results have been used in practice, this should be commented on and possibly supported by data on quality improvement, reduction in the number of breakouts, prolonging the life of the SEN, etc. The results of the simulations are useful from a qualitative point of view to get just a rough idea, but a quantitative comparison with reality is debatable. Steelmakers are interested in the influence of the position and shape of SEN on the quality of blooms, oscillation marks depth etc. and these effects are difficult to predict from velocity vector graphs calculated by a simplified numerical model. The authors predict or estimate the effects on quality, SEN life etc. Such conclusions should be compared and the model should be tuned according to the real process, not just the water model. The results give a simplified idea of the influence of analysed parameter while other parameters and quantities are constant which is far from reality. The authors should at least comment on this.

The text is written with little care. The paper contains many mistakes and inaccuracies. There are errors in names and symbols of quantities, units etc. There are also formal errors in the text, e.g. missing spaces, some units are in parenthesis, some not, etc. Texts describing the flow character and other results are too lengthy, some statements are similarly repeated in several places.

Comments:

row 54. In Fig. 1, some edges are bold and others are thin.

row 65. The thickness of the casting shell at a depth of 900 mm can reach several cm. Doesn't it really affect the flow of liquid steel?

row 80. Units (m/s) should belong to velocity, not to both uj and xj. xj is coordinate (m), not „direction“.

row 83. Here i and j are correctly called „direction“, units m/s are not in parenthesis, unlike the other units, applies to the whole article.

row 84. Missing spaces before units (Pa) and (Pa.s), applies to the whole article. It should be written uniformly and systematically. p is pressure rather than stress.

row 87. As citations equations (1) to (7) are not provided, it would be laborious to derive and check the equations, but e.g. in the equation (3) ∂xj is missing. The form of writing the equations is rather „economical“, more advanced equation editor should be used instead.

row 93. „Coefficient“ is a quantity by which another quantity is multiplied. µ and µt are should not be named „coefficients“.

row 94. σe is not explained, but σk is there twice.

row 94. Are all the empirical constants dimensionless or their dimensions are missing?

row 100. Start the line with lowercase w after the equation. There are redundant parentheses are in the equation.

row 101. (Kg) should be written with lowercase (kg). Units (J/kg) are for specific enthalpy, enthalpy is in (J).

row 102. Units (W/m.K) are used for thermal conductivity, not heat transfer coefficient. With respect to the equation in the row 100, sT and st,T cannot be dimensionless; units are missing.

row 105. Assume that the forces are in (N), then the equation (6) does not apply dimensionally.

row 108. Missing dimensions of forces FD and Fi. What is „the other force“? What is dimension of CD?

row 109. (i=x, y, z) is type mismatch, i means direction (dimensionless), while x, y, z are coordinates (m), should be written as a list or set i={x, y, z} or similarly.

row 158. Figure 2 is not commented in the text. What is PrePDF etc.?

row 159. Units (k) in the Table 2 should be capital (K).

row 160. Table 3. Correctly should be “Specific heat capacity” in units J/(kg.K), not in (J/kg).

Heat transfer coefficient is usually in units W/(m2.K), not in (W/m). The quantity is probably heat conductivity, but the units are still invalid.

row 163. What means „the prototype“? It is mentioned in the introduction that the article concerns a real casting machine.

rows 167, 169. Reynolds number vs. Reynolds Number

row 180. According to Fig. 3 it looks more like a slide valve then the rotameter.

row 185. None of the numbers in the equations (9), (10) and (11) are dimensionless.

row 189. Table 4. Is really mould of diameter 600 mm 900 mm long? It is usually shorter.

How steel-slag and water-oil interfacial tension coefficients have been determined?

row 193. The citation from which the equation (12) is taken should be provided.

row 204. Citation of the source of parameters should be provided.

row 214. „values of liquid surface fluctuation”? Fluctuation is a process, not a quantity. FR and FL is steel level height or oscillation amplitude or what exactly?

row 236. Instead of “second cool” use “secondary cooling zone”.

row 238. What means „the heat transfer of the mold fluxes“? Heat transfer from the liquid level? This is negligible in comparison to heat transfer to the mold wall.

row 300. The values on the horizontal axis of the chart are not in %. The name of the quantity on the axis is not „Percentage“ but e.g. „Removal rate“ or similar.

rows 314 – 317. What is a difference between the center line and center axis? By the way, axis is usually in the center.

row 369. Fig. 12. Vertical axis: Wrong units of pressure (N).

row 432. The term „impact depth„ should be explained at the place of its first occurrence, i.e. line 230.

row 504. The values on the horizontal axis of the chart are not in %. The name of the quantity on the axis is not „Percentage“ but „Removal rate“ or similar.

row 509. What is the uncertainty of the results being reported in hundredths of a percent?

row 526 – 536. It is not known what thermal boundary condition was specified on the mold surface. In fact, it is not constant and it depends on many parameters. Without measurements of temperature field in a real mold, the results are unpredictable. Obtained results have no real basis, just illustrative meaning.

row 540. The physical model is isothermal in contrast to the numerical model. The physical model cannot replace the verification of the numerical model. It allowed only a qualitative and very rough assessment of the nature of the flow.

Author Response

Response to Reviewer 1 Comments

Point 1: In the paper, the flow from submerged entry nozzle (SEN) into the mould is clearly elaborated by the method of numerical simulations. 

Response 1: Thank you, Sir.

Point 2: The article lacks a more detailed literary search focused on the effects of the quantities that are analyzed in the article, on the real casting process and quality.

Response 2: We find that references [9,10] described the negative effects of asymmetrical flow in mould, and references[6,7] provided the suppression of asymmetrical flow in mould by swirling nozzle. So we revised our article as: “The operation results in uneven distribution of the flow field, a large surface wave and uneven thickness of the solidified shell[9,10]. It was found that the swirling flow field driven by swirling nozzle could inhibit the asymmetrical flow in mould and could suppress the negative effects on the surface and/or internal quality of strand caused by asymmetrical flow[6,7].”rows41-44

Point 3: From the beginning, it is not explicitly stated whether the model calculates coupled flowing and cooling. The reader learns about it gradually and the information is inaccurate and incomplete. Thermal boundary conditions are not exactly described, even the most important boundary condition of heat flux into the mould wall.

Response 3: In our model the energy transfer equation was coupled with the flow field equations, and we add a comments of the couple of energy transfer equation with momentum and continuity equations. rows 79-80,100. The boundary condition of heat flux to the mould wall was added: 745kW/m2. row136

Point 4: The physical model is only isothermal. Either the two models cannot be compared, or it must be proved that the effect of steel cooling on the flow is negligible.

Response 4: The reference 10 proved that the effect of steel cooling on the flow is negligible. The results from water model was used to confirm the CFD results.

Point 5: Although the results are interesting, they largely prove what is already known or what can be expected. The article has mainly theoretical or academic value and demonstrates mastering of the use of commercial software. The real conditions are much more complicated. On the steel level, the interface between the casting powder and the molten steel changes dynamically mainly due to the oscillation of the mould and due to the interventions of the steel level control system. The interface between the real solid shell and the liquid phase is not a smooth cylinder; the shell thickness is not uniform etc.

Response 5: The target continuous caster we simulated was used to cast the low cost strands with different sizes. There are several steel mills producing strands with different sectional sizes of round strand with the eccentric SEN, but there were some quality issues caused by asymmetrical flow in mould, but the quality are not key point for low cost round bloom.The bosses of the steel mill wanted a solution to suppress the asymmetrical flow in mould with the eccentric SEN to obtain the balance between the qualtiy and cost, and we tell them the feasible solution through the simulation. They was satisfied with our solution, but they did not want to open up their products data in our article, as the data refered to some classified technologies.

Point 6: It is not clear, whether the results have been used in practice, this should be commented on and possibly supported by data on quality improvement, reduction in the number of breakouts, prolonging the life of the SEN, etc. The results of the simulations are useful from a qualitative point of view to get just a rough idea, but a quantitative comparison with reality is debatable. Steelmakers are interested in the influence of the position and shape of SEN on the quality of blooms, oscillation marks depth etc. and these effects are difficult to predict from velocity vector graphs calculated by a simplified numerical model. The authors predict or estimate the effects on quality, SEN life etc. Such conclusions should be compared and the model should be tuned according to the real process, not just the water model. The results give a simplified idea of the influence of analysed parameter while other parameters and quantities are constant which is far from reality. The authors should at least comment on this.

Response 6: Figure 20 (row 535) and rows 542-543 give the comments of the improved effects with optimized swirling nozzle.

Point 7: The text is written with little care. The paper contains many mistakes and inaccuracies. There are errors in names and symbols of quantities, units etc. There are also formal errors in the text, e.g. missing spaces, some units are in parenthesis, some not, etc. Texts describing the flow character and other results are too lengthy, some statements are similarly repeated in several places.

Response 7: Sorry for our carelessness, Sir.

Point 8: In Fig. 1, some edges are bold and others are thin.

Response 8: The bold edges are coated layers to prevent nozzle clogging and slag corrosion.

Point 9: The thickness of the casting shell at a depth of 900 mm can reach several cm. Doesn't it really affect the flow of liquid steel?

Response 9: As the velocity of molten steel at  a depth of 900 mm was less than 0.1m/s, the error of the shell thickness(~ 12mm) could be negligible.

Point 10: row 80. Units (m/s) should belong to velocity, not to both uj and xj. xj is coordinate (m), not „direction“.

Response 10: “In the formula, uj and xj represent the velocity component (m/s) and direction component;” was revised to “In the formula, uj represent the velocity component (m/s); xj is coordinate (m);” row83.

Point 11: row 83. Here i and j are correctly called „direction“, units m/s are not in parenthesis, unlike the other units, applies to the whole article.

Response 11: Revised to “where ui and uj denote the velocity of i and j direction (m/s); xi and xj stand for coordinates of the i and j directions (m);” rows86-87.

Point 12: row 84. Missing spaces before units (Pa) and (Pa.s), applies to the whole article. It should be written uniformly and systematically. p is pressure rather than stress.

Response 12: Revised to “p is pressure (Pa)” row87.

Point 12: row 84. Missing spaces before units (Pa) and (Pa.s), applies to the whole article. It should be written uniformly and systematically. p is pressure rather than stress.

Response 12: Revised to “p is pressure (Pa); μeff indicates the coefficient of effective viscosity (Pa·s) ” row87.

Point 13:row 87. As citations equations (1) to (7) are not provided, it would be laborious to derive and check the equations, but e.g. in the equation (3) ∂xj is missing. The form of writing the equations is rather „economical“, more advanced equation editor should be used instead.

Response 13: Revised to “∂(ρk)/∂t - ∂(ρkuj)/∂xj = ∂[(μt/σk - μ)(∂k/∂xi)]/∂xj + μt(∂uj/∂xi)(∂ui/∂xj + ∂uj/∂xi) - ρε ” between rows90 and 91.

Point 14: row 93. „Coefficient“ is a quantity by which another quantity is multiplied. µ and µt are should not be named „coefficients“.

Response 14: Revised to “In the formula, μt represents the turbulent viscosity , Pa·s; μ is the laminar viscosity , Pa·s;” row 96.

Point 15: row 94. σe is not explained, but σk is there twice.

Response 15: Revised to “ σε and σk are empirical constants, C1=1.43, C2=1.93, Cμ=0.09, σε=1.0, σk=1.43.” row 97.

Point 16: row 94. σe is not explained, but σk is there twice.

Response 16: Revised to “ σε and σk are empirical constants, C1=1.43, C2=1.93, Cμ=0.09, σk=1.0, σε=1.43.” row 97.

Point 17: Are all the empirical constants dimensionless or their dimensions are missing?

Response 17: Yes, they are dimensionless constants.

Point 18: row 100. Start the row with lowercase w after the equation. There are redundant parentheses are in the equation.

Response 19: Revised to “where keff can be calculated by equation keff= μ/σT+μt/σt,T ;”.

Point 19: row 101. (Kg) should be written with lowercase (kg). Units (J/kg) are for specific enthalpy, enthalpy is in (J).

Response 19: Revised to “the specific enthalpy of molten steel(J/kg); ”.

Point 20: row 102. Units (W/m.K) are used for thermal conductivity, not heat transfer coefficient. With respect to the equation in the row 100, sT and st,T cannot be dimensionless; units are missing.

Response 20: Revised to “ H represents the specific enthalpy of molten steel (J/kg); T stands for molten steel temperature (K); and keff is the effective thermal conductivity (W/m/K); σT=1.00 (K·s2/m2); σt,T=0.9 (K·s2/m2). ”Rows 103-105.

Point 21: row 105. Assume that the forces are in (N), then the equation (6) does not apply dimensionally.

Response 21: FD(u - up) is the drag force per particle mass; FD is the momentum exchange coefficient of drag force (1/s); Fi is the liquid inertia force per particle mass acting on particle as particle accelerating (i=x, y, z) (m/s2)

Point 22: row 108. Missing dimensions of forces FD and Fi. What is „the other force“? What is dimension of CD?

Response 22: Revised to “ u and up are the velocity of the fluid phase and the particle phase (m/s); μi is the kinetic viscosity of the fluid (Pa·s); ρ and ρp are the density of the fluid phase and the particle phase (kg/m3); dp is the particle diameter (m); Re is the Reynolds number of the particle; FD is the momentum exchange coefficient of drag force (1/s); Fi is the liquid inertia force per particle mass acting on particle as particle accelerating (i={x, y, z}) (m/s2);”. Fi is the liquid inertia force per particle mass acting on particle as particle accelerating (i=x, y, z) (m/s2), CD is a dimensionless parameter. Rows 109-113.

Point 23: row 109. (i=x, y, z) is type mismatch, i means direction (dimensionless), while x, y, z are coordinates (m), should be written as a list or set i={x, y, z} or similarly.

Response 23: Revised to “ Fi is the liquid inertia force per particle mass acting on particle as particle accelerating (i={x, y, z}) (m/s2);” rows 112-113.

Point 24: row 158. Figure 2 is not commented in the text. What is PrePDF etc.?

Response 24: PrePDF is a combustion model’s computing step in Fluent software. As we did not use the combustion model in this article, the PrePDF was removed from Figure2. A comment was added for Figure 2 at Row 162. Figure 2 Revised as follow:

Point 25: Units (k) in the Table 2 should be capital (K).

Response 25: Units (k) in the Table 2 was revised to capital (K).

Point 26: row 160. Table 3. Correctly should be “Specific heat capacity” in units J/(kg.K), not in (J/kg).

Response 26: row 160. Table 3. Correctly was revised to “Specific heat capacity”

Point 27: Heat transfer coefficient is usually in units W/(m2.K), not in (W/m). The quantity is probably heat conductivity, but the units are still invalid.

Response 27: The unit of heat transfer coefficient was revised to units W/(m2.K) row 165

Point 28: What means „the prototype“? It is mentioned in the introduction that the article concerns a real casting machine.

Response 28: „the prototype“ means the actual mould in caster.

Point 29: rows 167, 169. Reynolds number vs. Reynolds Number

Response 29: Reynolds number was revised to Reynolds number. Row 169

Point 30: row 180. According to Fig. 3 it looks more like a slide valve then the rotameter.

Response 30: The rotameter was revised to slide gate.

Point 31: row 185. None of the numbers in the equations (9), (10) and (11) are dimensionless..

Response 31: row 190: The numbers were revised to

Re = uodo/ν

(9)

Fr = uo2/gdo

(10)

We = ρuo2do/σ

(11)

Point 32: row 189. Table 4. Is really mould of diameter 600 mm 900 mm long? It is usually shorter.

Response 32: The target mould we simulated is really 900mm long.

Point 33: How steel-slag and water-oil interfacial tension coefficients have been determined?

Response 33: The steel-slag tension coefficient and water-oil tension coefficient were cited from references.

Point 34: row 193. The citation from which the equation (12) is taken should be provided.

Response 34: The citation of equation (12) was add. Row 199.

Point 35: row 204. Citation of the source of parameters should be provided.

Response 35: The citation of the source of parameters was provided.

Point 36: row 204. Citation of the source of parameters should be provided.

Response 36: The citation of the source of parameters was provided.

Point 37: row 214. „values of liquid surface fluctuation”? Fluctuation is a process, not a quantity. FR and FL is steel level height or oscillation amplitude or what exactly?

Response 37: FR and FL is steel level height amplitude measured by wave gauges.

Point 38: row 236. Instead of “second cool” use “secondary cooling zone”.

Response 38: row 241. Instead of “second cool” was revised to “secondary cooling zone”

Point 39: row 238. What means „the heat transfer of the mold fluxes“? Heat transfer from the liquid level? This is negligible in comparison to heat transfer to the mold wall.

Response 39: row 243. was revised to “the heating of the mold flux”, that is upward flow bringing more heat to the mold flux.

Point 40: row 300. The values on the horizontal axis of the chart are not in %. The name of the quantity on the axis is not „Percentage“ but e.g. „Removal rate“ or similar.

Response 40: The x-axis of fig. 8 was revised to “Removal rate (%)”

Point 41: rows 314 – 317. What is a difference between the center row and center axis? By the way, axis is usually in the center.

Response 41: rows 319-320. was revised to “Some representative positions were selected to analyze the temperature distribution, including the the radial temperature distribution of liquid level, radial temperature distribution of mold outlet plane (Z=900mm), the center axis temperature distribution of the mold and the center axis of the SEN.”. rows 329-333. was revised to “As shown in Figure 9, when the SEN is align with the mold, the distribution of temperature field in the mold is symmetrical at both sides of the mold or the SEN. When the molten steel is poured under SEN eccentric conditions, the radial temperature distribution of the mold is obviously one side high and one side low. The symmetry of the temperature distribution on the liquid surface and the outlet plane (Z=900mm) of the mold is disturbed.”.

Point 42: row 369. Fig. 12. Vertical axis: Wrong units of pressure (N).

Response 42: Fig.12 was revised as follow.

Point 43: row 432. The term „impact depth„ should be explained at the place of its first occurrence, i.e. row 230.

Response 43: row 435 was revised to “The side-hole jets and central stream have comprehensively effect on impact depth.”. And row 234 was revised to “Its velocity gradually decreases with the increase of the impact depth, the shortest distance from the liquid surface of the mold to a certain position, in which the average downward velocity of the fluid on the cross section of the mold is equal to the casting speed, is defined as the impact depth.”.

Point 44: row 504. The values on the horizontal axis of the chart are not in %. The name of the quantity on the axis is not „Percentage“ but „Removal rate“ or similar.

Response 44: row 435: Fig.18(a) was revised as follow:

Point 45: row 509. What is the uncertainty of the results being reported in hundredths of a percent?

Response 45: row 512 was revised to “However, under the SEN eccentric conditions, the overall removal efficiency of inclusions in the mold still decreased from 51.7% to 50.36%, and the difference is only 2.6%.”

Point 46: row 526 – 536. It is not known what thermal boundary condition was specified on the mold surface. In fact, it is not constant and it depends on many parameters. Without measurements of temperature field in a real mold, the results are unpredictable. Obtained results have no real basis, just illustrative meaning.

Response 46: Dear Sir, our meaning is that the swirling flow caused by swirlling nozzle could relieve the asymmetric flow in mould driven by eccentric locating of SEN, and the reduction of asymmetric flow may relieve the uneven level of shell thickness in turn. We think your advice is more reasonable, we just provide a weak discuss. Thanks.

Point 47: row 540. The physical model is isothermal in contrast to the numerical model. The physical model cannot replace the verification of the numerical model. It allowed only a qualitative and very rough assessment of the nature of the flow.

Response 47: Thank you for your wonderful comments. We really learn a lot.

Reviewer 4 Report

A nice piece of work. Some minor corrections in English language typos are suggested:

Line 164: “it is necessary that Reynolds number, the Froude number and Webber number of the prototype and the model are equal,”, suggested: “it is necessary that the Reynolds, Froude, and Webber numbers for the prototype and model are equal,”

Line 167-168: “are far more than 105, which belong to”, suggested: “are by far more than 105, belonging to”

Line 171: “when the Froude number and Weber number are equal” suggested: “when the Froude and Weber numbers are equal”

Line 251: “is gradually is” suggested: “is gradually”

Line 263: “fluxes involved” suggested: “fluxes infiltration”

Line 285: “This is mainly because the” suggested: “This is mainly due to”

Line 308: “not to float upward” suggested: “not float upward”

Line 308: “inside of round bloom” suggested: “inside the round bloom”

Line 346: “of the mold;” suggested: “of the mold”

Lines 398, 402, & 448: “° ,” suggested: “°,”

Line 410: “have been done non-dimensional treatment.” Suggested: “have become dimensionless.”

Line 461: “SEN has similar” suggested: “SEN has a similar”

Line 494: “velocity of slag” suggested: “velocity for slag”

Line 527: “by more than 10 K. which m can” suggested: “by more than 10 K which can”

Author Response

Thank you for your hard work for revising of our article. Thanks again.

Point 1: Line 164: “it is necessary that Reynolds number, the Froude number and Webber number of the prototype and the model are equal,”, suggested: “it is necessary that the Reynolds, Froude, and Webber numbers for the prototype and model are equal,” 

Response 1: Line 164 is corrected to “it is necessary that the Reynolds, Froude, and Webber numbers for the prototype and model are equal,”. Thanks a lot.

Point 2: Line 167-168: “are far more than 105, which belong to”, suggested: “are by far more than 105, belonging to”

Response 2: Line 168-169 is corrected to “are by far more than 105, belonging to”.

Point 3: Line 171: “when the Froude number and Weber number are equal” suggested: “when the Froude and Weber numbers are equal”

Response 3: Line 172 is corrected to “when the Froude and Weber numbers are equal”.

Point 4: Line 251: “is gradually is” suggested: “is gradually”

Response 4: Line 252-253 is corrected to “is gradually replaced”.

Point 5: Line 263: “fluxes involved” suggested: “fluxes infiltration”

Response 5: Line 264 is corrected to “fluxes infiltration”.

Point 6: Line 285: “This is mainly because the” suggested: “This is mainly due to”

Response 6: Line 286 is corrected to“This is mainly due to”.

Point 7: Line 308: “not to float upward” suggested: “not float upward”

Response 7: Line 309 is corrected to “not float upward”.

Point 8: Line 308:“inside of round bloom” suggested: “inside the round bloom”

Response 8: Line 309 is corrected to “inside the round bloom”.

Point 9: Line 346:“of the mold;” suggested: “of the mold”

Response 9: Line 347: “of the diameter of the mold” is corrected to “the mold diameter”.

Point 10: Lines 398, 402, & 448: “° ,” suggested: “°,”

Response 10: Lines 399, 403, & 449: “° ,” is corrected to “°,”

Point 11: Line 410: “have been done non-dimensional treatment.” Suggested: “have become dimensionless.”

Response 11: Line 411: “have been done non-dimensional treatment.” is corrected to “have become dimensionless.”

Point 12: Line 461: “SEN has similar” suggested: “SEN has a similar”

Response 12: Line 461: “SEN has similar” is corrected to “SEN has a similar”

Point 13: Line 494: “velocity of slag” suggested: “velocity for slag”

Response 13: Line 495: “velocity of slag” suggested: “velocity for slag”

Point 14: Line 527: “by more than 10 K. which m can” suggested: “by more than 10 K which can”

Response 14: Line 527: “by more than 10 K. which m can” is corrected to “by more than 10 K which can”

Round 2

Reviewer 3 Report

Dear authors,

Thank you for the explanation. I accept the answers to most of my comments. Please make some remaining corrections.

Point 3.

Although the model is greatly simplified, it can be accepted but it should be always commented on. It must be defended or demonstrated that the simplification does not significantly affect the quantity under investigation. For example, the previously missing and added surface condition of the heat flux on the surface of the mold (row 136) which has been set to be constant over the entire mold surface, is very far from reality where it differs from the meniscus to the end of the mold several times.

I understand that managers from plants often do not want to share the real effects of the research.

Point 7.

There are still quite a few formal mistakes in the article, such as missing spaces in the text, redundant parentheses in formulas or missing parenthesis (e.g. see units at the row 122 and many others). I assume that everything will be fine in the final version of the text after formatting.

Point 9.

OK. Such facts should be commented in a scientific paper.

Point 22.

It is common to write „CD (1)“ or sometimes „CD (-)“ in case of dimensionless quantities. It depends on requirements of the journal editor. Should be written „kinematic viskosity“ instead of „kinetic viscosity“.

Point 26.

If the value =41 is heat conductivity, the units should be (W/(m.K)).

Point 29.

There is another "Reynolds Number" with capital "N" in the line 174.

Point 40.

Still the values on the horizontal axis are not in %. They should be multiplied by 100.

Point 44. See Point 40, the same error.

Author Response

Response to Reviewer 3 Comments(second round)

Point 3: Although the model is greatly simplified, it can be accepted but it should be always commented on. It must be defended or demonstrated that the simplification does not significantly affect the quantity under investigation. For example, the previously missing and added surface condition of the heat flux on the surface of the mold (row 136) which has been set to be constant over the entire mold surface, is very far from reality where it differs from the meniscus to the end of the mold several times.

I understand that managers from plants often do not want to share the real effects of the research. 

Response 3: The average heat flux was measured through the difference between in and out temperature of the mould cooling water by us at the operating mould in the actual round bloom caster. A heat flux distribution curve was used in our model. As the poor enghlish level, the thermal boundary condition was only described as the average heat flux. Rows 137-138 was revised to “, and the height distance from the meniscus level dependant heat flux distributionon with the average heat flux(745 kW/m2) is employed as the heat transfer condition.”.

Point 7: There are still quite a few formal mistakes in the article, such as missing spaces in the text, redundant parentheses in formulas or missing parenthesis (e.g. see units at the row 122 and many others). I assume that everything will be fine in the final version of the text after formatting. 

Response 7: Rows82-190 was checked and revised for these issues.

Point 9: OK. Such facts should be commented in a scientific paper. 

Response 9: Thank you for your reasonable and helpful suggestion, it is really helpful, Thanks again.

Point 22: It is common to write „CD (1)“ or sometimes „CD (-)“ in case of dimensionless quantities. It depends on requirements of the journal editor. Should be written „kinematic viskosity“ instead of „kinetic viscosity 

Response 22: Row 113 was revised to CD is the drag force coefficient (-).. Row 106 was revised to dynamic viscosity.

Point 26: If the value =41 is heat conductivity, the units should be (W/(m.K)). 

Response 26: Row 166 was revised to Effective thermal conductivity (W/(m·K)).  

Point 29: There is another "Reynolds Number" with capital "N" in the line 174. 

Response 29: Row 175 was revised to fluid is not affected by Reynolds number,.  

Point 40 and 44: Still the values on the horizontal axis are not in %. They should be multiplied by 100. 

Response 40 and 44: Fig.8, Fig. 11, Fig.14(b) and Fig.18(a) was revised.  
